# Strategies to identify and suppress crosstalk signals in DEER experiments with Gd(III) and nitroxide spin-labeled compounds

Markus Teucher[1], Mian Qi[2], Ninive Cati[2], Henrik Hintz[2], Adelheid Godt[2], and Enrica Bordignon[1]

[1]Faculty of Chemistry and Biochemistry, Ruhr University Bochum, Universitätsstraße 150, 44801 Bochum, Germany
[2]Faculty of Chemistry and Center for Molecular Materials (CM$_2$), Bielefeld University, Universitätsstraße 25, 33615 Bielefeld, Germany

**Correspondence:** Enrica Bordignon (enrica.bordignon@rub.de)

**Abstract.** DEER spectroscopy applied to orthogonally spin-labeled biomolecular complexes allows to simplify the assignment of intra- and inter-molecular distances, thereby increasing the information content per sample. In fact, various spin labels can be addressed independently in DEER experiments due to spectroscopically non-overlapping central transitions, distinct relaxation times and/or transition moments, hence they are referred to as spectroscopically "orthogonal". Molecular complexes which are, for example, orthogonally spin-labeled with nitroxide (NO) and gadolinium (Gd) labels give access to three distinct DEER "channels", optimized to selectively probe NO-NO, NO-Gd and Gd-Gd distances. Nevertheless, it has been previously recognized that crosstalk signals between individual DEER channels can occur, for example, when a Gd-Gd distance appears in a DEER channel optimized to detect NO-Gd distances. This is caused by residual spectral overlap between NO and Gd spins, which therefore, cannot be considered as perfectly 'orthogonal'. Here, we present a systematic study on how to identify and suppress crosstalk signals that can appear in DEER experiments using mixtures of NO-NO, NO-Gd and Gd-Gd molecular rulers, characterized by distinct, non-overlapping distance distributions. This study will help to correctly assign the distance peaks in homo- and hetero-complexes of biomolecules carrying non-perfectly orthogonal spin labels.

## 1 Introduction

### 1.1 DEER

Double Electron-Electron Resonance (DEER, also known as PELDOR) is an electron paramagnetic resonance (EPR) pulsed dipolar spectroscopy (PDS) technique introduced by Milov et al. (Milov et al., 1981, 1984) and further developed by Spiess and Jeschke (Martin et al., 1998; Pannier et al., 2000) that probes the $r^{-3}$-dependent dipolar coupling interaction between adjacent unpaired electron spins. In general, DEER allows the extraction of precise distance information on spin-labeled biomolecules from 1.5 nm to 6-8 nm, but the upper limit can be extended up to 16 nm (Schmidt et al., 2016) for perdeuterated samples. DEER is an established technique in structural biology (Jeschke, 2012, 2018), complementary to X-ray crystallography, NMR spectroscopy and cryo electron microscopy. Perspectively, it is seen among the most promising methods for in-cell studies (Plitzko et al., 2017).

DEER is usually performed using the dead-time free 4-pulse sequence (Martin et al., 1998; Pannier et al., 2000), a two-frequency experiment that allows detecting the dipolar modulation of the observer echo induced by changing the position of the pump pulse within the dipolar evolution time. The primary DEER trace contains an inter-molecular background function that needs to be fitted and separated from the desired intra-molecular dipolar signal.

A reliable fit of the background function relies on recording the primary DEER time trace as long as possible, so that the background decay function is visible after the dipolar oscillations have decayed. This is usually difficult to experimentally achieve for distances $> 5\text{-}6\,\text{nm}$, especially for samples carrying low concentrations of fast relaxing spins, as it is the case e.g. for spin-labeled membrane proteins. Decreasing the spin concentration alleviates the background problem, because at concentrations $< 10\,\mu\text{M}$ the background is an almost flat function, which is easier to be fitted and removed from the trace. Ambiguous background fitting can cause large uncertainties in distance distributions, that can be quantified by data validation approaches available in most software packages like DeerAnalysis (Jeschke et al., 2006) or LongDistances (Altenbach, 2020).

Removing the fitted background function from the primary DEER time trace (Ibáñez and Jeschke, 2020) results in the form factor that can be fitted using several approaches, most prominently Tikhonov regularization (Chiang et al., 2005; Jeschke et al., 2006; Edwards and Stoll, 2018) or Gaussian fitting (Brandon et al., 2012; Stein et al., 2015), yielding the distance distribution between intra-molecular dipolarly coupled spins. The recently introduced neural network analysis of DEER data (Worswick et al., 2018) allows direct analysis of primary DEER time traces, providing distance distributions with an uncertainty estimate based on variations in the fits of multiple networks.

## 1.2 "Orthogonal" spin labeling

In multispin systems carrying the same type of spin label, the assignment of distances within the overall distance distribution can be challenging due to the presence of ghost peaks (Jeschke et al., 2009; von Hagens et al., 2013), the suppression of long distances (Junk et al., 2011; Ackermann et al., 2017) and the intrinsic difficulties in disentangling multiple distance contributions, often already when only three spin labels are present in the system (Jeschke et al., 2009; Pribitzer et al., 2017). However, the analysis is simplified for oligomeric systems with a defined symmetry (Valera et al., 2016).

Orthogonal spin labeling (introduced by (Lueders et al., 2011; Kaminker et al., 2012; Yulikov et al., 2012)) facilitates the assignment of distances via selectively addressable DEER channels that give access to distance information of specific spin pairs at a time, thereby increasing the information content that can be obtained from a single sample (reviewed in (Yulikov, 2015)). In fact, two distinguishable spin labels in a system give access to three DEER channels: two channels probing the interactions among the labels of the same type and one channel probing interactions between the two different label types. Depending on the system under study, signals can appear in none, one, two or all three DEER channels. The term orthogonal refers to spin labels that are spectroscopically distinguishable from each other and that can be addressed and/or detected independently, e.g. via distinct resonance frequencies, relaxation behavior or transition moments. Despite most spin labels are non-perfectly orthogonal, it was shown that specific inter-spin interactions can be addressed independently, as demonstrated by several publications on a large number of combinations of spin labels, e.g. nitroxides in combination with trityl (Shevelev et al., 2015; Joseph et al., 2016; Jassoy et al., 2017), Gd[III] (Lueders et al., 2011; Kaminker et al., 2012; Yulikov et al., 2012;

Lueders et al., 2013; Garbuio et al., 2013; Kaminker et al., 2013; Gmeiner et al., 2017a, b; Teucher et al., 2019; Shah et al., 2019; Galazzo et al., 2020), Fe[III] (Ezhevskaya et al., 2013; Abdullin et al., 2015; Motion et al., 2016), Cu[II] (Narr et al., 2002; Bode et al., 2008, 2009; Meyer et al., 2016) or Mn[II] (Kaminker et al., 2015; Akhmetzyanov et al., 2015; Meyer and Schiemann, 2016). The orthogonal spin labeling approach has also been extended to more than two orthogonal spin labels (Wu et al., 2017).

In case of a non-negligible spectral overlap of the orthogonal labels, crosstalk signals between the DEER channels might appear depending on the degree of orthogonality between the labels and their relative abundance within the sample. This issue has already been addressed in some studies in literature before (Gmeiner et al., 2017a; Wu et al., 2017; Shah et al., 2019; Teucher et al., 2019), but was not yet systematically investigated.

## 1.3 The combination of nitroxide and gadolinium[III] spin labels

Nitroxides (NO) and Gd[III]-based spin labels (Gd) are fairly common for DEER experiments on biomolecules. Nitroxides are $S = 1/2$ spin systems with a spectral width in the order of $10\,\text{mT}$ at Q band ($\approx 35\,\text{GHz}$). Gd[III]-based spin labels are $S = 7/2$ systems extending over $450\,\text{mT}$ at Q band with a sharp central $|-1/2\rangle \rightarrow |+1/2\rangle$ transition whose maximum is usually about $10.4\,\text{mT}$ ($\approx 291\,\text{MHz}$) higher in magnetic field than the maximum of the NO spectrum. The two spins can be selectively addressed because of their different transition moments (Schweiger and Jeschke, 2001). In fact, a $\pi$-pulse for NO corresponds to a $4\,\pi$-pulse for the $|-1/2\rangle \rightarrow |+1/2\rangle$ transition of Gd (Yulikov, 2015), which stands for a $12\,\text{dB}$ difference in applied microwave power. Additionally, NO and Gd have distinct $T_1$ relaxation times, therefore, by using short shot repetition times (srt) it is possible to saturate the slow relaxing NO signal at 10 K and enhance the contribution of the Gd signal in the observer echo in DEER (Lueders et al., 2011; Kaminker et al., 2012).

In this work, we focus on three-channel DEER experiments performed at Q band using mixtures of NO and Gd spin labels. These two spin probes give access to three DEER channels, hereafter referred to as: NONO, NOGd and GdGd. We chose three rulers, namely an NO-NO, an NO-Gd and a Gd-Gd ruler with distinct non-overlapping distance distributions to study in a systematic way the signals in all detectable DEER channels if one, two or three different rulers are present in the same sample at different stoichiometric ratios. We characterize ruler combinations and DEER channels that are prone to crosstalk signals, quantify their relative strengths and provide methods to identify and suppress the unwanted contributions.

## 2 Materials and methods

### 2.1 Samples

In this work, we utilized the Gd-Gd ruler $Na_2[\{Gd^{III}(PyMTA)\}\text{-}(EP)_5E\text{-}\{Gd^{III}(PyMTA)\}]$ (Qi et al., 2016a), the NO-Gd ruler $Na[\{Gd^{III}(PyMTA)\}\text{-}(EP)_2\text{-}NO\bullet]$ (Ritsch et al., 2019), and the NO-NO ruler $\bullet ON\text{-}(EP)_2P\text{-}NO\bullet$ (for structural formulae see Fig. 1). In these compounds two $\{Gd^{III}(PyMTA)\}^-$ (Qi et al., 2016b) complexes, a $\{Gd^{III}(PyMTA)\}^-$ complex and a nitroxide, and two nitroxides are held by a rod-like spacer at a distance of 4.7 nm, 2.5 nm, and 2.0 nm, respectively. Because of their geometry and the rather high stiffness of the spacer (Jeschke et al., 2010) their interspin distances are well-defined. All

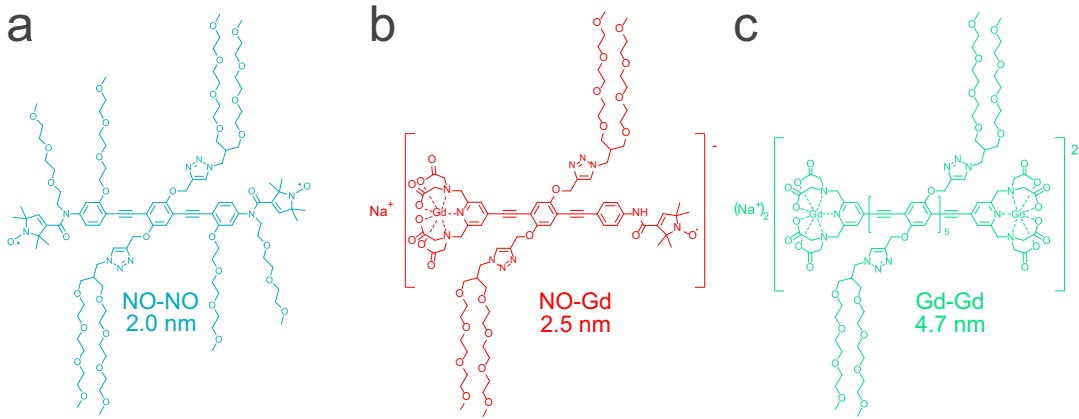

**Figure 1.** Structural formulae of the NO-NO, NO-Gd and Gd-Gd rulers. Indicated are the experimentally detected mean distances between the paramagnetic centers.

rulers are water soluble and can therefore be detected in the same environment as water soluble proteins. The synthesis and characterization of the Gd-Gd and the NO-Gd rulers was published before (Qi et al., 2016a; Ritsch et al., 2019), while the synthesis of the water soluble NO-NO ruler is described in the SI Part A.

The DEER samples were prepared using stock solutions of the rulers in $H_2O$ at concentrations of 50 - 100 μM. To each sample 50% v/v deuterated glycerol was added as cryoprotectant yielding the final spin concentrations given in Table S1 (SI Part B). 40 μl of each sample were inserted into 3 mm outer diameter quartz tubes and shock frozen in liquid nitrogen.

### 2.2 Instrumentation

#### 2.2.1 Spectrometers

Continous wave (cw) EPR experiments for NO spin counting were performed at X band using a MiniScope MS 5000 spectrometer (Magnettech by Freiberg Instruments). All pulsed EPR experiments were performed using a Bruker Biospin Q-band Elexsys E580 spectrometer equipped with a 150 W TWT amplifier from Applied Systems Engineering and a Bruker SpinJet-AWG (±400 MHz bandwidth, 1.6 GSa/s sampling rate, 14 bit amplitude resolution) in combination with a home-made Q-band resonator for 3 mm sample tubes (Tschaggelar et al., 2009; Polyhach et al., 2012).

#### 2.2.2 Transient nutation experiments

Nutation experiments were performed using the sequence (nutation pulse)-(1000 ns)-(π/2)-(400 ns)-(π)-(400 ns)-(echo) with 16 ns for the Gaussian π/2-pulse and 32 ns for the Gaussian π-pulse. The nutation pulse length was incremented starting from 0 in 2 ns steps and the position of the detection pulses as well as of the acquisition trigger was displaced using the same increment. For the data shown in Fig. 2, the frequency was placed in the center of the resonator dip and the amplitudes of all pulses were changed from 100 to 10% keeping the main attenuator at 0, 6, or 12 dB. The intensity of the echo (single point

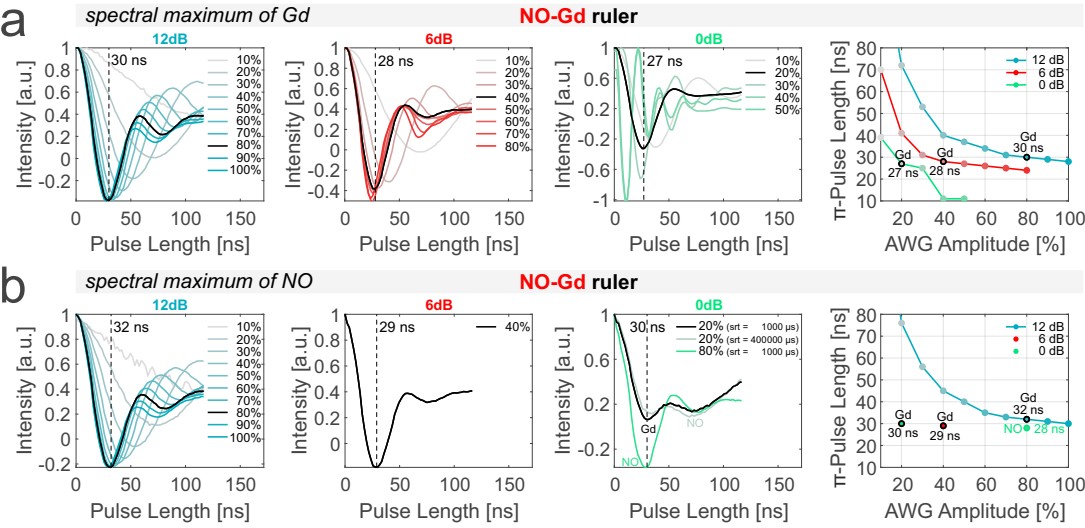

**Figure 2.** Power dependence of NO and Gd $\pi$-pulses. Transient nutation experiments were performed at different spectral positions of the NO-Gd ruler at 10 K. Data were recorded at 12, 6 and 0 dB attenuation (columns 1 to 3), varying the AWG amplitude. The fourth column shows the correlation of the extracted $\pi$-pulse lengths (first minimum of the nutation transient) and the AWG pulse amplitude (in %) for the different main attenuator settings (in dB). All extracted $\pi$-pulse lengths are given in Table S2 (SI Part B). The transients shown in (a) were recorded on the spectral maximum of the Gd and in (b) on the maximum of the NO-spectrum which is 10.4 mT lower in magnetic field than the maximum of the Gd for the utilized sample (see Fig. 3) using a shot repetition time (srt) of 1000 $\mu$s (if not stated differently). (a) In our setup, 12 dB attenuation and 80% AWG amplitude correspond to a 30 ns Gaussian $\pi$-pulse on Gd. Doubling the power (6 dB) always requires halving the AWG amplitude (highlighted in black for a $\approx$ 30 ns $\pi$-pulse). (b) A srt of 1000 $\mu$s makes the nutation experiments more sensitive to Gd at 10 K (see relaxation data given in Fig. S1 to S3 and Table S3 to S4, SI Part B). At 0 dB main attenuation and 20% AWG amplitude the nutation of the NO becomes also visible (black). Prolonging the srt to 400,000 $\mu$s at 20% amplitude slightly increases the amplitude of the NO nutation with respect to the nutation of Gd (gray). A pulse amplitude of 80% gives also 30 ns $\pi$-pulse length (green), which corresponds to a $\pi$-pulse on the NO spins.

detection on the maximum) was recorded versus the nutation pulse length and the position of the first minimum of the nutation transient was taken as the $\pi$-pulse length (Fig. 2, first three columns). The rightmost column in Fig. 2 shows the correlation between the amplitude of the AWG-pulses (expressed in %) and the $\pi$-pulse length for the different main attenuator settings (in dB). All experiments were performed using the NO-Gd ruler, placing the field either at the maximum of the Gd signal
($|-1/2\rangle \rightarrow |+1/2\rangle$ transition) or at the spectral maximum of the NO (Fig. 2(a) and (b)). The analysis of the data shows that at 12 dB attenuation, a 30 ns $\pi$ pulse can be obtained at 80% pulse amplitude, while at 6 dB approximately 40%, and at 0 dB about 20% pulse amplitude are required to obtain 28 and 27 ns $\pi$ pulses, respectively. This demonstrates a linear dependency between $B_1$ and AWG amplitude. In fact, doubling $B_1$ from 12 to 6 dB ($-6$ dB attenuation in power) requires a decrease of
the AWG amplitude from 80% to 40% and an increase in $B_1$ by a factor of 4 (corresponding to $-12$ dB) correlates with a change in the AWG amplitude from 80% to 20%. The linear dependence between the AWG amplitude and the intensity of

the transiently recorded pulses in transmission mode (TM) was previously shown (SI of (Teucher and Bordignon, 2018)), and here we demonstrate that this linearity is maintained also after the TWT (Traveling-Wave Tube) amplification up to 70-80% AWG amplitude (at this input power the TWT starts to saturate). The same trend could be detected at the maximum position of the NO spectrum, using a shot repetition time (srt) of 1000 μs, which saturates the NO signal and thereby enhances the Gd signal (Fig. 2(b)). Therefore, we can conclude that we sample mostly the Gd $|-1/2\rangle \rightarrow |+1/2\rangle$ transition even at 10.4 mT lower in field with respect to the maximum of the Gd spectrum. To address the complications arising from the overlap of the NO and Gd signals (see Fig. 3), we performed nutation experiments at 0 dB and 20% AWG amplitude on the maximum of the NO spectrum using different srt values (Fig. 2(b), third column). Using a fast srt of 1000 μs, two minima of nearly equal intensity are detected in contrast to the single minimum for the nutation performed on the maximum of the Gd using the same parameters. These minima are created by the superposition of the nutations of the Gd (first minimum at 30 ns) and the NO spins (second minimum). Because of the slow $T_1$-relaxation of the NO at 10 K, performing the same experiment at srt 400,000 μs increases the contribution of the nitroxide-related minimum. When the AWG amplitude is set to 80% (which corresponds to a 12 dB increase in power with respect to the 20% amplitude), we detected a first minimum at 30 ns, which is attributable to a π-pulse for the NO spins. In fact, at this power the π-pulse for the Gd spins should be ≈ 10 ns (see Fig. 2(a)). Therefore, changing the microwave power by 12 dB (e.g. from 80 to 20% AWG amplitude) allows to selectively address either Gd or NO spins. Overall, the nutation experiments allow a precise determination of the optimal length of the π-pulses for NO and Gd in all DEER setups.

### 2.2.3 DEER setup

DEER experiments were performed using the dead-time free 4-pulse sequence $(\pi/2)_{obs} - (d_1) - (\pi)_{obs} - (d_1 + T) - (\pi)_{pump} - (d_2 - T) - (\pi)_{obs} - (d_2) - (echo)$ (Martin et al., 1998; Pannier et al., 2000) with 16-step phase cycling (Tait and Stoll, 2016) using (0)-(π) for $(\pi/2)_{obs}$ and $(\pi)_{obs}$, and (0)-(π/2)-(π)-(3π/2) for $(\pi)_{pump}$. All pulse experiments were performed using monochromatic pulses with a Gaussian amplitude modulation function, predefined as pulse shape 1 in Bruker Xepr 2.6b.119. In Xepr, the pulse length $t_p$ of a Gaussian pulse is defined as its time base (truncation at 2.2% of its maximum amplitude) which is related to its full width at half maximum (FWHM) by $t_p = 2\sqrt{2ln2} \cdot \text{FWHM} \approx 2.3548 \cdot \text{FWHM}$ (Teucher and Bordignon, 2018). Gaussian π/2- and π-pulses at the observer frequency were created by varying the pulse amplitude at a fixed pulse length to maintain a uniform excitation bandwidth for the refocused echo (Teucher and Bordignon, 2018). The length of the Gaussian pulses was optimized individually for each experiment via transient nutation experiments, as shown in Fig. 2.

In all DEER experiments, the main frequency of the microwave bridge was set to the observer position with the AWG synthesizing the frequency offset required for the pump pulse. More details about the utilized three-channel DEER setups are given in Fig. 3. The evaluation of the DEER data was performed with DeerAnalysis2019 using the Gaussian fitting routine assuming a homogeneous 3D background function and the neural network analysis (DeerNet) (Jeschke et al., 2006; Worswick et al., 2018) to obtain an error estimation. Gaussian fitting was chosen over Tikhonov regularization since it simplifies data evaluation for distributions with well-defined distance peaks, allows simultaneous fitting of components with very different distribution widths and enables quantification of the relative contributions of the distance peaks, which is optimal for the

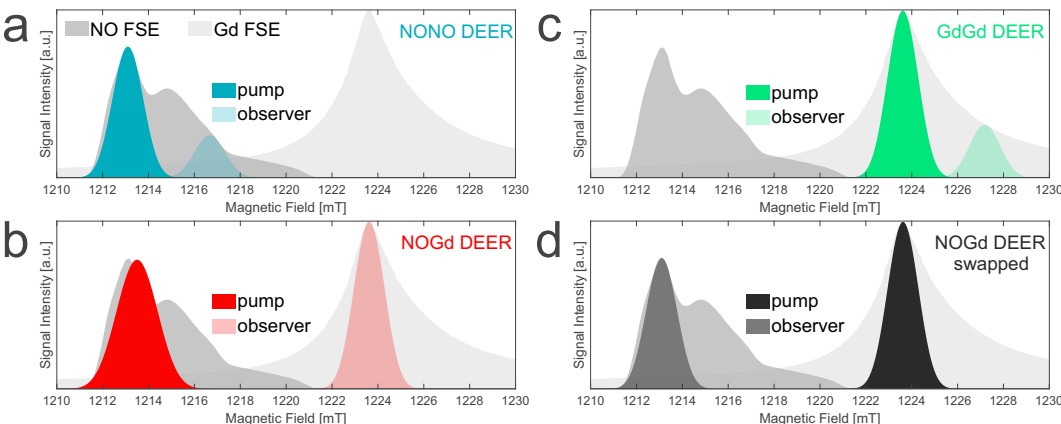

**Figure 3.** Three-channel DEER setups. Two field-swept echo (FSE) spectra of NO and Gd are represented as shaded gray areas, superimposed at their relative spectral positions. Gaussian π-pulse excitation profiles are shown at the respective pump and observer positions, simulated with EasySpin 5.2.2 (Stoll and Schweiger, 2006) using the provided functions "pulse" and "exciteprofile". The excitation profiles represent ideal Gaussian π-pulses, without taking into account the spectral shape, the non-linearity of the signal response, the resonator profile and the Q factor. Therefore they are only indicative for the excitation bandwidth of the pulses. Nevertheless, we found that the correct choice of the pulse lengths and a positioning of the pump and observer frequencies in such a way that the two simulated excitation profiles do not overlap allows to experimentally minimize the "2+1" signal at the end of DEER traces (Teucher and Bordignon, 2018). In setups (a-c), Gaussian observer pulses of 32 ns time base length (13.6 ns FWHM) for π/2 and π (Teucher and Bordignon, 2018) were used in combination with a shot repetition time (srt) of 1000 μs. (a) NONO DEER: 32 ns Gaussian pump at the spectral maximum of NO; observer pulses 100 MHz lower in frequency; pump/observer placed symmetrical in resonator profile; performed at 50 K. (b) NOGd DEER: 32 ns Gaussian observer pulse at the spectral maximum of Gd; Gaussian pump pulse of 24 ns (10.2 ns FWHM) placed in the center of the resonator profile (minimum possible pulse length in our setup) 280 MHz higher in frequency than the observer; performed at 10 K. (c) GdGd DEER: as in (a), except for the pump pulse placed on the maximum of the Gd spectrum; performed at 10 K. (d) Swapped NOGd DEER setup: 32 ns Gaussian observer pulses at the spectral maximum of NO; 32 ns Gaussian pump pulse 291 MHz lower in frequency than the observer; performed at 30 K with an srt of 10,000 μs. Observer placed +50 MHz off-center in the resonator profile.

analysis performed. However, a comparison of Gaussian and Tikhonov analysis can be found in Fig. S4 (SI Part B). Both approaches are in a good agreement with respect to each other and also with respect to the DeerNet analysis, although DeerNet was trained using only NONO DEER data.

### 2.2.4 Relaxation measurements

Longitudinal ($T_1$) and transverse ($T_m$) relaxation measurements were performed at 10, (30) and 50 K on all samples at different spectral positions in correspondence with the pump/observer positions of the DEER setups introduced in Fig. 3. The relaxation data are shown in Fig. S1 to S3 and Table S3 to S4 (SI Part B).

   $T_1$ was measured using the inversion recovery sequence (π)-(T)-(π/2)-(180 ns)-(π)-(180 ns)-(echo) with a 32 ns Gaussian inversion π-pulse separated by a variable time T from the 16-32 ns Gaussian echo sequence. The signal was recorded by

160 integrating over the FWHM of the echo (= 32 ns) and plotting the echo intensity versus T. The initial time T was set to 800 ns and incremented in N·ΔT steps. The $T_1$ values were extracted in MATLAB assuming a Bloch model for relaxation. The fully recovered magnetization was normalized to one and $T_1$ ($= T_{1\,[0.26]}$) was extracted as the time where the echo intensity reaches a value of 0.26 according to:

$$I(t) = 1 \left( 1 - 2\,e^{-\frac{T}{T_1}} \right) \tag{1}$$

with $T = T_1$

$$I(t) = 1 \left( 1 - \frac{2}{e} \right) \approx 0.26 \tag{2}$$

Based on the small variations observed in technical repeats, we estimate an error of 5%. For $T_{1\,[0.26]}$ values longer than 0.35 ms the error increases to 20% due to the limited length (3.5 ms) of the inversion recovery traces detected (due to AWG memory limitations).

$T_m$ was measured using the echo decay sequence (π/2)-(T)-(π)-(T)-(echo) with 16-32 ns Gaussian pulses separated by a variable time T. The signal was acquired by integrating over the FWHM of the echo (= 32 ns) with an initial interpulse delay T of 180 ns which was incremented in N·ΔT steps. The echo intensity was plotted versus the interpulse delay T. As commonly reported in literature (Shah et al., 2019), the $T_m$ ($= T_{m\,[10\%]}$) values were extracted from the echo decay curves as the time T at which the echo intensity is decayed to 10% of its original value using MATLAB. Based on technical repeats, we estimate an 175 error of 10%.

## 3 Experimental results and discussion

### 3.1 Isolated rulers

The DEER characterization of the three individual rulers is shown in Fig. 4. Since the NO-NO and the Gd-Gd rulers contain only one type of label, we probed only one DEER channel per sample, namely the NONO or GdGd channel, respectively. For 180 the NO-Gd ruler we probed all three DEER channels. The dipolar frequencies, distance distributions and modulation depths obtained on the isolated rulers are characteristic sample- and setup-dependent parameters which will be used in the following to identify and quantify crosstalk signals in the ruler mixtures. An overview of all DEER data and the quantification of the fractions of each distance peak in the overall distribution is given in Table S5 (SI Part B).

The NONO DEER time trace (blue) detected on the NO-NO ruler in Fig. 4(a) shows a dipolar frequency with a 35% 185 modulation depth, corresponding to a well-defined 2 nm distance. The GdGd DEER time trace (green) detected on the Gd-Gd ruler shows a dipolar frequency with a modulation depth of ≈ 3%, corresponding to a monomodal distance distribution centered at 4.7 nm (see Fig. 4(b)). The uncertainties in the distance distributions for both rulers are are negligible, as shown by the neural network analysis presented in Fig. 4(b,c). The Tikhonov analysis is shown in Fig. S4 (SI Part B).

The time traces obtained on the NO-Gd ruler with the three DEER channels are shown in Fig. 4(c). The NOGd DEER 190 time trace (red) shows a defined dipolar frequency (30% modulation depth) correlated with a 2.5 nm distance. The distance

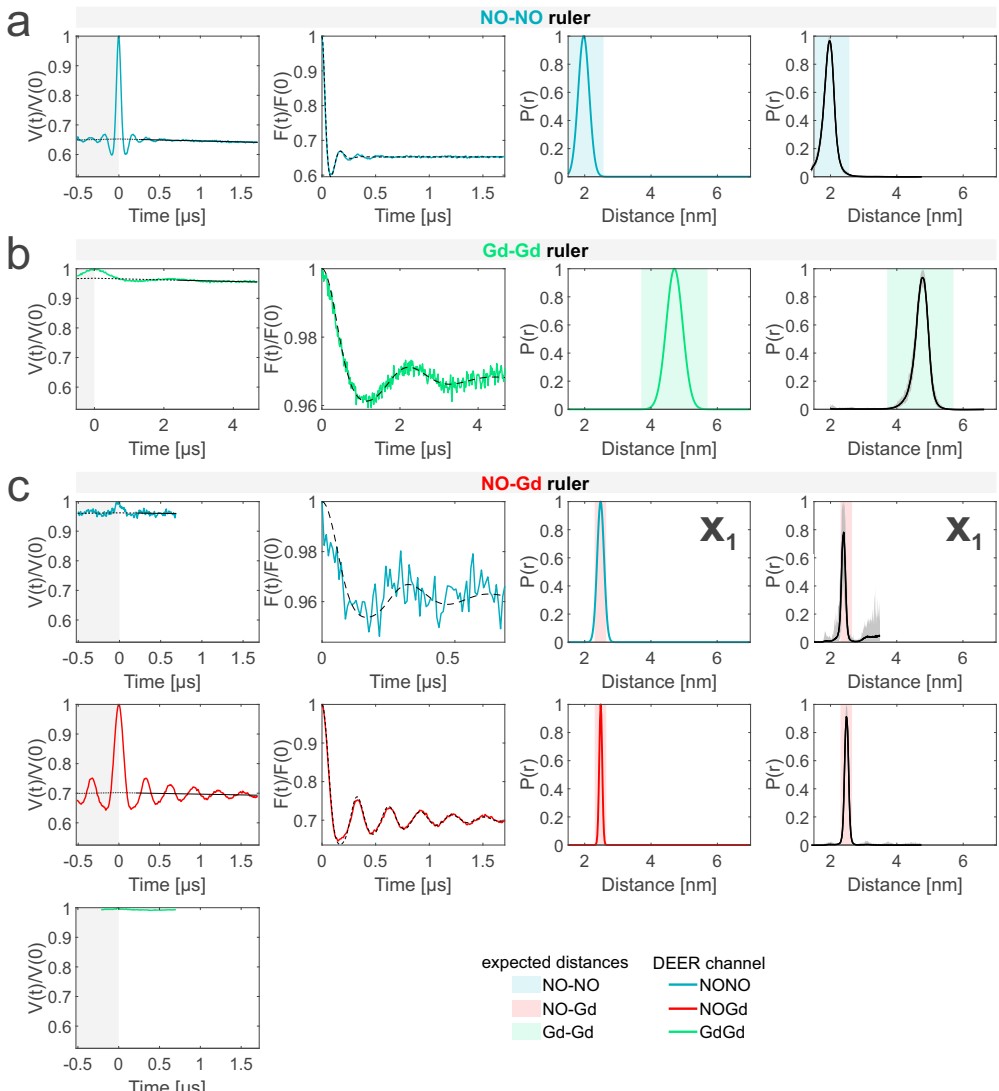

**Figure 4.** Characterization of the isolated rulers. The DEER setups are introduced in Fig. 3. First column, primary data with background fit (gray areas are excluded from data evaluation); second column, form factors (obtained by dividing the primary data by the background function) with fit from Gaussian fitting routine; third column, obtained distance distributions; fourth column, DeerNet analysis (Generic network) to provide an error estimation. A Tikhonov analysis of the data is shown in Fig. S4 (SI Part B). An overview over all DEER data is given in Table S5 (SI Part B). The time traces, form factors and distance distributions recorded with the NONO DEER channel are colored in blue, those recorded with the GdGd channel are colored in green, and those recorded with the NOGd channel are colored in red. Regions in which distances can be expected based on the rulers present in the specific sample are represented as shaded blue, green and red areas in the distance distributions. "$X_1$" is a NO-Gd crosstalk in the NONO DEER channel.

obtained via neural network analysis is consistent and shows negligible uncertainties. However, Tikhonov analysis extracted an additional peak of low intensity centered at 2 nm, which could be consistent with minor orientation selection effects (Fig. S4 (SI Part B)). Unexpectedly, the NONO DEER channel (blue) also contains a dipolar signal with 4% modulation depth whose distance distribution coincides with the one obtained in the NOGd DEER channel. This is a DEER channel crosstalk signal, caused by the unintended excitation of spectrally overlapping Gd spins via the pump and/or observer pulses. Due to the fast phase memory time of the nitroxide spins in the NO-Gd ruler at 50 K (see Fig. S2 and Table S4, SI Part B), only a noisy 1 μs trace could be detected, and as a consequence, the neural network analysis provides larger uncertainties in the main distance peak. Such a crosstalk signal is significant, because its $\approx 4\%$ modulation depth is in the order of 10% of the maximally achievable modulation depth for the spin-labeled NO-NO ruler (see Fig. 4(a)). We classify this signal as a NO-Gd crosstalk in the NONO DEER channel and designate it as $\mathbf{X}_1$. The GdGd channel (green) shows no dipolar modulation, confirming that the NO-Gd ruler is monomeric in solution and that the signal detected in the NONO DEER channel is indeed a crosstalk signal between DEER channels.

## 3.2 Ruler mixtures

In this section we investigate the appearance of crosstalk signals between the DEER channels in samples containing mixtures of the three rulers. We chose to analyze two different molar ratios to address the effect of relative spin concentrations on the strength of the crosstalk signals. The data with a 2-fold excess of the Gd-Gd rulers with respect to the others is presented in the main text, while we show a full data set of the rulers in equimolar mixtures in the SI (Table S6 and Figs. S5-S7, SI Part B). The reproducibility of the data presented are shown with independent repetitions performed on the isolated rulers in Fig. S5 and on the mixtures of two rulers in Fig. S6.

The three DEER experiments performed on the mixture of the NO-NO ruler with the NO-Gd ruler in a 1:1 molar ratio are shown in Fig. 5. The NONO DEER channel contains the expected distance distribution of the isolated NO-NO ruler characterized in Fig. 4(a). The NOGd channel reproduces the signal obtained on the isolated NO-Gd ruler previously shown in Fig. 4(b). The GdGd channel shows no dipolar modulation, in line with the absence of Gd-Gd rulers in this sample.

The NO-Gd crosstalk signal previously detected in the NONO channel ($\mathbf{X}_1$) for the isolated NO-Gd ruler in the mixture of the NO-NO with the NO-Gd rulers in Fig. 4(c) is not experimentally resolved (see Fig. 5). If we consider that the NO spins of the NO-Gd ruler are observed and the Gd spins are partially excited by the pump pulse, we suggest that the absence of this crosstalk signal is due to the fact that the NO spins of the NO-Gd ruler have a shorter phase memory time $T_m$ than those in the NO-NO ruler at 50 K ($T_m \approx 2$ μs versus 4.6 μs, see Fig. S2 and Table S4, SI Part B), which strongly decreases their contribution in the observer echo for the detected 2 μs time trace. Additionally, in this sample, only 1/3 of the NO observer signal in the NONO channel originates from the NO-Gd ruler, which will further decrease the modulation depth of the crosstalk signal with respect to the case in which only the NO-Gd ruler is present (see Fig. 4(c)). If we consider that the Gd spins are partially observed and the NO spins are pumped, the presence of the NO-NO ruler reduces the relative contribution of the Gd spins in the observer echo, thereby decreasing the modulation depth of the crosstalk signal. Accordingly, in this mixture the NO-Gd crosstalk signal is negligible and only the dominant signal contribution at 2 nm arising from the NO-NO ruler is detectable.

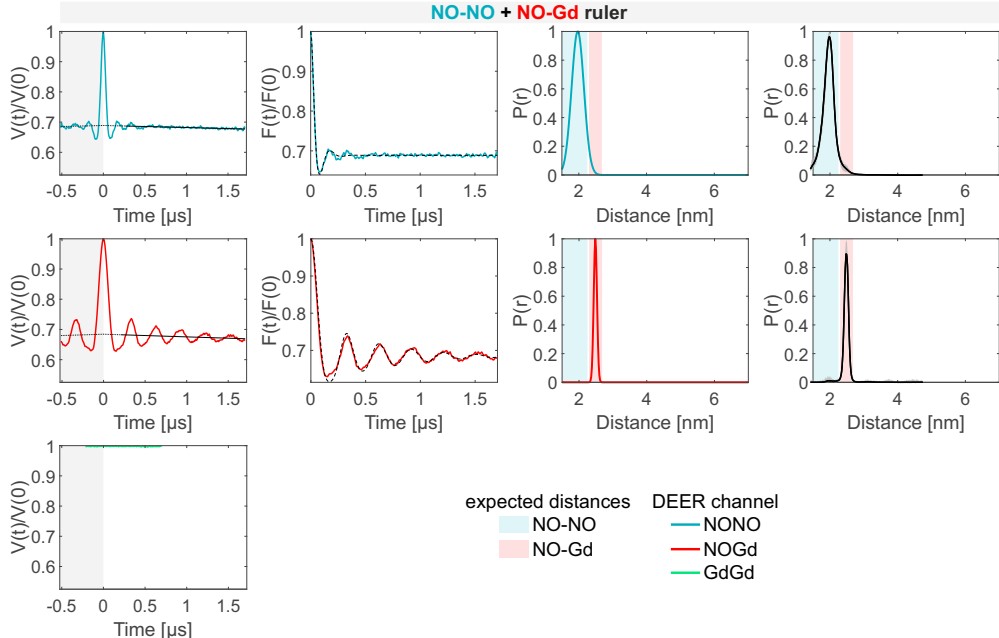

**Figure 5.** Sample containing the NO-NO and the NO-Gd rulers mixed in a 1:1 molar ratio. The legend is the same as in Fig. 4. No crosstalk signals are detected in this sample.

The analysis of the sample containing the NO-NO and the Gd-Gd ruler in a 1:2 or 1:1 molar ratio is presented in Fig. 6 and Fig. S6(b) (SI Part B), respectively. No differences could be observed when different ratios were used. Both the NONO and the GdGd channels reproduce nicely the DEER signals obtained from the isolated NO-NO and Gd-Gd rulers. As there is no NO-Gd ruler present in the sample, no signal would be expected in the respective DEER channel. However, a dipolar frequency was detected with a 4% modulation depth, which is attributed to a Gd-Gd crosstalk signal in the NOGd channel (defined as $X_2$), as shown by the fit performed with a single Gaussian centered at the same mean distance as that of the isolated Gd-Gd ruler (see Fig. 6, solid line). The neural network analysis revealed a second distance peak centered at 3.5 nm (highlighted with an asterisk), which can also be found when fitting the data with two Gaussian peaks, which improves the root mean square deviation between fit and data (Fig. 6, broken line). To understand the origin of the peak at 3.5 nm, a series of DEER experiments using a stock solution of Gd-maleimide DOTA was performed. It was found that the 3.5 nm peak arises from a sinusoidal signal with a frequency of $\approx 1$ MHz which is independent on the chosen srt. This signal has no dipolar origin, we can exclude that it is an ESEEM effect, and it appears also in $MnCl_2$ solutions. We could remove it only by decreasing the power of the pump pulse to zero (more information in Fig. S8, SI Part B). Therefore, we assign the 3.5 nm peak to an artifact in our setup. The strength of this artifact varies in different measurements and it is mostly visible when traces with small modulation depths and high signal-to-noise are detected.

The results of the experiments with the 1:2 mixture of the NO-Gd ruler with the Gd-Gd ruler are presented in Fig. 7. The NONO DEER channel of this sample shows the NO-Gd crosstalk signal in the NONO DEER channel ($X_1$) as reported for the

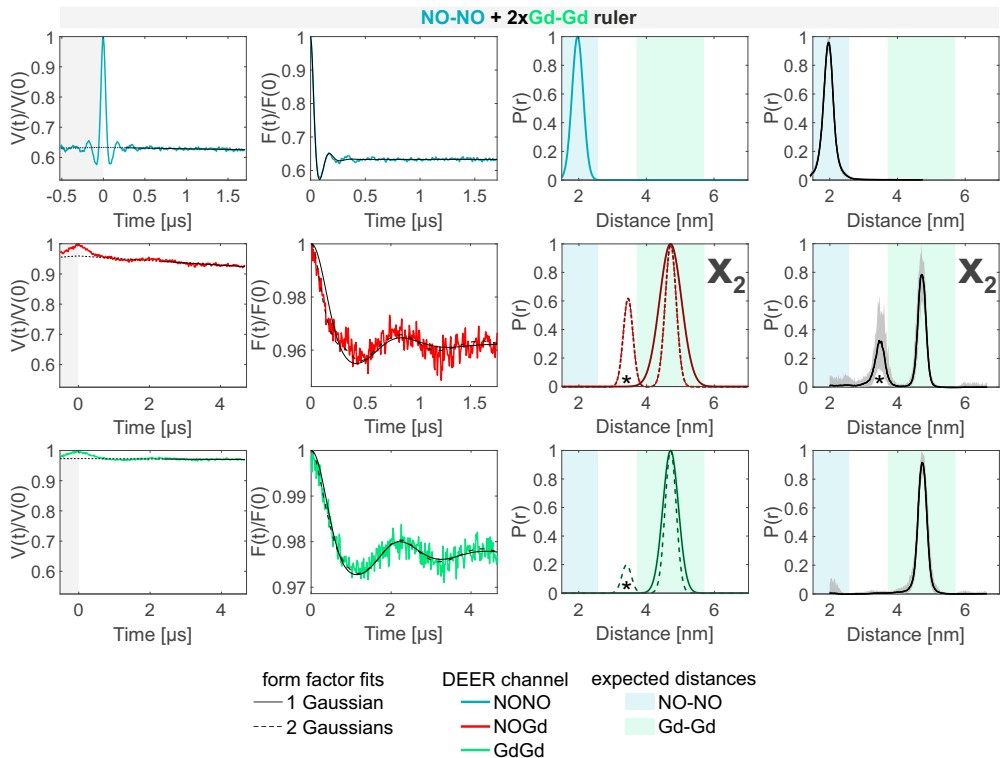

**Figure 6.** Sample containing the NO-NO and the Gd-Gd rulers mixed in a 1:2 molar ratio. The legend is the same as in Fig. 4. The form factors in the NOGd and in the GdGd channel were fitted using both a single Gaussian (solid line) and two Gaussians (broken line) to highlight the appearance of a spectrometer-specific artifact signal corresponding to a 3.5 nm distance (highlighted with an asterisk, see Fig. S8, SI Part B). The NOGd DEER channel contains a Gd-Gd crosstalk signal in absence of a NO-Gd distance designated as $\mathbf{X_2}$. A comparison on how different neural networks fit this crosstalk signal is shown in Fig. S9(a) (SI Part B).

isolated NO-Gd ruler in Fig. 4(c). Likewise, just a short time trace could be recorded due to the fast phase memory time of the NO spins in the NO-Gd ruler (see Fig. S2 and Table S4, SI Part B). The presence of this crosstalk signal corroborates our interpretation that it can be detected only in the absence of extra NO spins in the sample. The GdGd channel in our DEER

setup (pump at the maximum of the spectrum and observer at a higher field, see Fig. 3(c)), is intrinsically crosstalk-free and shows the expected pure Gd-Gd distance. In contrast, the NOGd channel contains, besides the expected NO-Gd distance, a Gd-Gd crosstalk signal defined as $\mathbf{X_3}$ which is fully resolved in the 4.7 µs time trace presented in Fig. 7.

However, this crosstalk signal could not be identified in the 1:1 mixture, indicating that the relative concentration of the Gd-Gd ruler modulates the intensity of such unwanted signal in the NOGd channel (see Fig. 7 versus Fig. S6(c), SI Part B).

The crosstalk signals identified in the NOGd DEER channel in Fig. 6 and Fig. 7 are both Gd-Gd crosstalk signals in the NOGd channel, however, we decided to keep a distinction in the names based on the absence/presence of a "real" NO-Gd distance which will have an influence on the identification and suppression procedure discussed below.

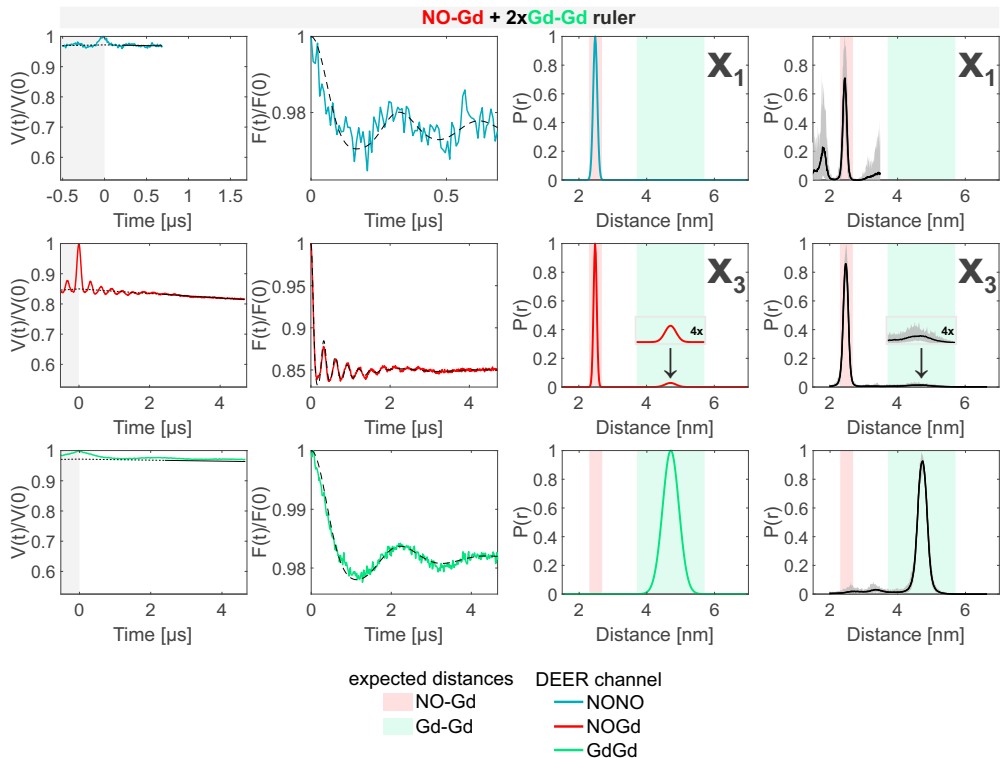

**Figure 7.** Sample containing the NO-Gd and the Gd-Gd rulers mixed in a 1:2 molar ratio. The legend is the same as in Fig. 4. The NONO DEER channel contains a NO-Gd crosstalk signal ($X_1$) and the NOGd channel contains a Gd-Gd crosstalk signal in presence of a NO-Gd distance designated as $X_3$. A comparison on how different neural networks fit this crosstalk signal is shown in Fig. S9(b) (SI Part B).

The DEER data obtained on the sample containing the NO-NO, NO-Gd and Gd-Gd rulers in a 1:1:2 ratio are presented in Fig. 8. Essentially, these data can be seen as a superposition of the data detected on the pairwise mixtures of rulers. The NONO
DEER channel shows the distance distribution of the NO-NO ruler but lacks the $X_1$ crosstalk signal due to the presence of additional NO signals. Besides the expected NO-Gd ruler distance, the NOGd channel shows the Gd-Gd crosstalk signal $X_3$ as in Fig. 7, which is clearly visible in the asymmetry of the time trace due to the underlying low frequency characteristic of the Gd-Gd dipolar function. Finally, the GdGd DEER channel resolves the Gd-Gd distance free of crosstalk signals.

In conclusion, we identified three non-negligible crosstalk signals in the NONO and NOGd DEER channels and we showed
that the GdGd DEER setup with the observer frequency placed on the maximum of the Gd signal and the pump frequency at the high field edge of the Gd spectrum (see Fig. 3(c)) is intrinsically crosstalk-free in all experimental conditions tested.

### 3.3    DEER channel crosstalk identification and suppression

The DEER channel crosstalk signals discussed in this work are named as follows: $X_1$ is a NO-Gd crosstalk signal in the NONO channel, while $X_2$ and $X_3$ are both Gd-Gd crosstalk signals in the NOGd channel but either in the absence or presence

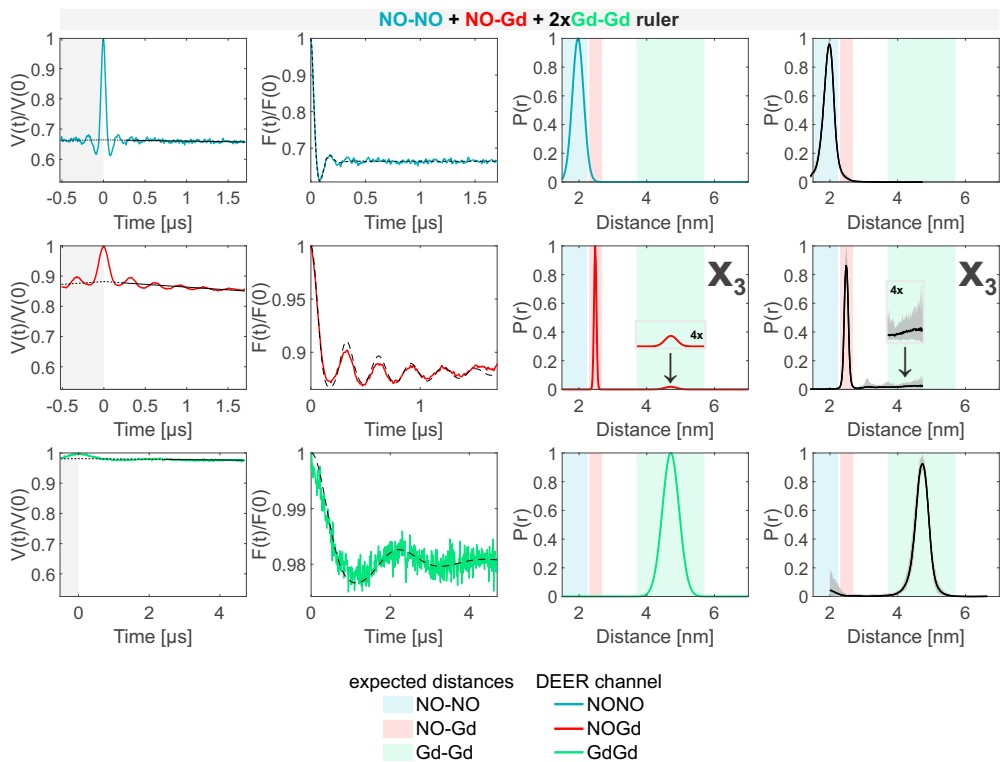

**Figure 8.** Sample containing the NO-NO, NO-Gd and Gd-Gd rulers mixed in a 1:1:2 ratio. The legend is the same as in Fig. 4. "$X_3$" is a Gd-Gd crosstalk signal in the NOGd DEER channel in presence of a NO-Gd distance.

of a "real" NO-Gd signal, respectively. An overview of all crosstalk signals that are possible to occur based on the spectral overlap between NO and Gd at the pump and/or observer positions and those experimentally detected with our sample/setup combination is presented in Table S7 (SI Part B).

The origin of the NO-Gd crosstalk signal in the NONO channel $X_1$ at 50 K (see Fig. 4(c) and 7) lies in the excitation of the Gd spins with a pump pulse close to $4\pi$ (see Fig. 2(b)) while optimally observing the NO spins and/or an excitation of the NO spins with an optimal pump $\pi$-pulse while sub-optimally observing the Gd spins. We could not find an experimental strategy to minimize this crosstalk signal. However, when the NO-NO ruler (therefore a real NO-NO distance) is present, the contribution of this unwanted signal was found to be negligible due to the shorter phase memory time of the NO in the NO-Gd ruler with respect of the NO in the NO-NO ruler, and because of the presence of an additional NO signal contribution in the observer echo which is not dipolarly coupled to the Gd spins (see Fig. 5 and 8).

Fig. 9(a,b) provides identification and suppression strategies for the $X_3$ crosstalk signal (Gd-Gd crosstalk in the NOGd channel in the presence of a real NO-Gd distance) from Fig. 7 and 8. In our NOGd DEER setup (see Fig. 3(b)), the observer pulses excite only Gd spins, therefore, the Gd-Gd crosstalk signal originates from sub-optimally pumping the Gd at the NO position due to the spectral overlap. In Fig. 9(a) we present a strategy to identify this crosstalk signal by lowering the power

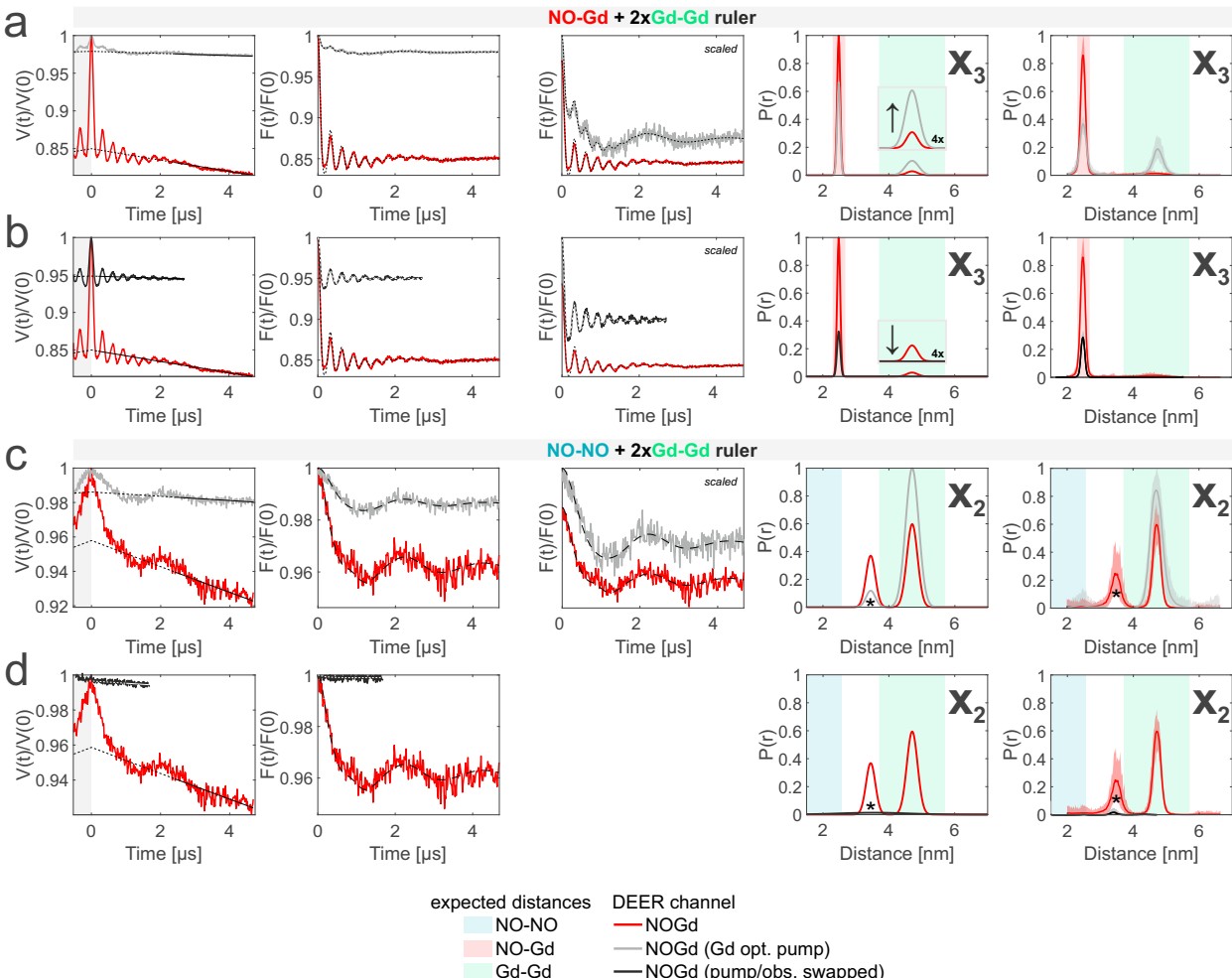

**Figure 9.** Crosstalk signal identification. First column, primary data with background fit (gray areas are excluded from data evaluation); second column, form factors with fit obtained with Gaussian fitting routine; third column, form factors scaled to same modulation depth and offsetted by constant value; fourth column, obtained distance distributions; fith column, DeerNet analysis (Generic network) to provide an error estimation. (a-b) Sample containing the NO-Gd and the Gd-Gd rulers mixed in a 1:2 molar ratio (related to Fig. 7). The NOGd channel contains a Gd-Gd crosstalk signal in presence of a NO-Gd distance ($X_3$). (a) Decreasing the pump pulse power in the standard NOGd DEER setup (see Fig. 3(b)) from optimally pumping NO (red) to optimally pumping the Gd (-12 dB, gray) changes the signal-to-crosstalk ratio and thereby allows to identify the crosstalk signal. (b) By pumping the Gd and observing on the NO (swapped NOGd DEER setup, see Fig. 3(d)) the Gd-Gd crosstalk signal can be fully removed from the NOGd DEER channel (black). (c-d) Sample containing the NO-NO and the Gd-Gd rulers mixed in a 1:2 molar ratio (related to Fig. 6). The NOGd DEER channel contains a Gd-Gd crosstalk signal in absence of a NO-Gd distance ($X_2$). The distance indicated with an asterisk originates from a spectrometer-specific artifact signal. (c) Analogous to (a). (d) Analogous to (b).

of the pump pulse at the NO position by 12 dB, in order to increase the contribution of the Gd-Gd dipolar frequency in the time trace and to simultaneously suppress the modulation depth of the NO-Gd frequencies. In red we present the NOGd DEER time trace with the distance distributions extracted by Gaussian fitting and DeerNet from Fig. 7 and in gray the time trace obtained with a pump pulse of 12 dB less power. Decreasing the power of the pump pulse decreases the modulation depth by a factor of 7 (from 15% to 2%) and changes the ratio of the modulation depths of the two dipolar frequencies (and of the extracted distance peaks) in favor of the crosstalk signal, as expected (see arrow in the inset). Therefore, the 12 dB decrease in power allows the identification of the $X_3$-crosstalk signal since it promotes the intensity of the Gd-Gd crosstalk distance while strongly decreasing the intensity of the NO-Gd distance. Notably, the Gd-optimized pump pulse with 12 dB less power still partially pumps the NO and therefore the DEER trace contains a residual NO-Gd signal contribution.

Fig. 9(b) presents a strategy to completely suppress this crosstalk signal by swapping the pump and observer positions in the NOGd channel (see Fig. 3(b) versus (d)). This strategy has already proven to be effective in a case study (Shah et al., 2019). Usually, the NO spins are pumped and the Gd spins are observed (see Fig. 3(b)) to optimize the modulation depth in the NOGd channel. If the positions of the pump and observer pulses are exchanged, the observer pulses are placed in the region of the spectral overlap at the spectral maximum of the NO, while the pump pulse will excite only Gd spins. The advantage of the swapped setup is that the observer sequence can act as a better filter for one spin species than a single pump pulse. The observer sequence uses $\pi/2$- and $\pi$-pulses optimized by nutation experiments for the NO signal, which will overflip the Gd spins, decreasing their contribution in the observer echo. The main disadvantage of this approach is the long shot repetition time required to observe on the NO (100 ms for the NO with respect to 1 ms for Gd in the conventional setup at 10 K), which makes DEER data acquisition impractically long. Additionally, the modulation depth will be smaller for the desired NO-Gd signal. The latter issue could be improved using frequency- and amplitude-modulated broadband pump pulses, as it was previously shown for Gd spin pairs (Doll et al., 2013; Spindler et al., 2013; Doll et al., 2015; Bahrenberg et al., 2017). In Fig. 9(b) we present in red the NOGd DEER time trace with the distance distributions extracted by Gaussian fitting and DeerNet from Fig. 7; in black the time trace obtained with a swapped NOGd DEER setup (see Fig. 3(d)) recorded at 30 K. To achieve a sufficient signal-to-noise ratio in a reasonable measuring time, we thought to increase the temperature from 10 to 50 K to shorten the longitudinal relaxation time of the NO, thereby enabling the use of a faster srt (in the order of 1 ms). To maintain the polarization introduced by the pump pulse, the longitudinal relaxation time of the pumped Gd spins needs to remain longer than the dipolar evolution time of the DEER sequence, but we found that at 50 K the $T_1$ of Gd is too short (see Table S3, SI Part B). The best compromise between the $T_1$ of the Gd and NO spins is at 30 K for the investigated samples (see Fig. S3, SI Part B). The time trace detected at 30 K (black in Fig. 9(b)) shows a dipolar frequency with a modulation depth of 5%, characteristic of the pure NOGd signal. Therefore, the GdGd crosstalk signal could be fully suppressed in the NOGd channel using the swapped setup maintaining a good signal-to-noise ratio.

In principle, it is possible to even better tune the power of the observer pulses to achieve an even more pronounced filtering of the Gd signals with respect to the NO signal, as shown in Fig. S10 (SI Part B) for the spectra detected at 10 K. In fact, we found that decreasing the pulse amplitude of the observer pulses to 50%, further increases the intensity of the NO signal in the

refocused echo with respect to the Gd signal, therefore further improving the selectivity of the observer sequence towards the NO.

In Fig. 9(c,d) we show the effects of the 12 dB power decrease in the pump (identification strategy) and of the swapped setup (suppression strategy) on the Gd-Gd crosstalk signal detected in the NOGd channel in the absence of a NO-Gd signal ($\mathbf{X}_2$) from Fig. 6. Decreasing the pump pulse power by 12 dB (gray trace in Fig. 9(c)) decreases the dipolar modulation of the Gd-Gd crosstalk signal (and diminishes the impact of the spectrometer artifact) with respect to the red trace (taken from Fig. 6). The modulation depth contribution of the Gd-Gd signal in this setup is about 1.25%, which is in line with the modulation

depth obtained with the same setup on the isolated Gd-Gd ruler under the same experimental conditions shown in Fig. S11 (SI Part B). Unfortunately, this is not a good strategy to identify such crosstalk signals. However, in line with the conclusion drawn above, using the swapped setup at 30 K removed the dipolar frequency of the Gd-Gd signal in the NOGd DEER channel, thereby suppressing the unwanted crosstalk signal.

## 4   Conclusions and outlook

In this work we thoroughly investigated the appearance of crosstalk signals between the three possible DEER channels at Q-band frequencies with mixtures of NO-NO, NO-Gd and Gd-Gd rulers with non-overlapping distance distributions.

     Our experimental findings further corroborate the notion that crosstalk signals can be expected in the NONO and NOGd DEER channels, which are performed with pump and/or observer pulses positioned in the region of the NO-Gd spectral overlap. In contrast, the GdGd DEER setup presented in Fig. 3(c) is intrinsically crosstalk-free. The detected crosstalk signals

are of experimental relevance when biomolecular complexes labeled with NO and Gd are investigated, since they are in the order of 10% of the maximally expected modulation depth in the respective DEER channel and therefore entail the risk of data misinterpretation when unknown mixtures of orthogonally-labeled proteins are studied. We also found that the relative strengths of the crosstalk signals depend on the relative molar ratio of the different types of spin labels. Notably, other factors such as relative labeling efficiencies, widths of the peaks in the distance distribution, presence of long distances close to

detection limit etc. may also modulate the relevance of the crosstalk signals in the overall data analysis in biological samples.

     The NO-Gd crosstalk in the NONO channel ($\mathbf{X}_1$) was found to be negligible if a real NO-NO dipolar oscillation is present, due to the dominating signal contribution from the NO spins that are not interacting with the present Gd spins and due to the larger modulation depth of the real signal (in the order of 30 - 40%) with respect to the 2 - 4% for the crosstalk signal. We were not able to find a suitable spectroscopic approach to identify or suppress the NO-Gd crosstalk signal in the NONO channel,

apart from an identification strategy based on the comparison of the distance distributions detected in the NONO and NOGd DEER channels on the same sample, which can be ambiguous. A possible strategy to clarify whether a crosstalk signal is detected, is to prepare an analogous sample with the Gd-labeled proteins exchanged with the unlabeled variants. If the NONO DEER channel is free of dipolar oscillations, the signal previously detected was a NOGd crosstalk signal; otherwise, if the same dipolar frequency is detected, it can be concluded that it is a real NO-NO distance.

We found that the Gd-Gd crosstalk signals in the NOGd DEER channel are the most relevant unwanted signals in terms of their relative modulation depths with respect to the desired NO-Gd signals. Therefore, their presence can be detrimental in the analysis of complex protein mixtures. We propose a quick identification strategy based on decreasing the power of the pump pulse positioned at the maximum of the nitroxide spectrum by 12 dB to optimally pump the Gd spins. This method changes the relative intensities of the GdGd and NOGd signals, thereby allows only the identification of $\mathbf{X}_3$. We show that Gd-Gd crosstalk

signals can be completely suppressed by swapping the position of the pump and observer pulses in the NOGd DEER channel at 30 K. The temperature was chosen to find the best compromise between a sufficiently short $T_1$ of the NO spins (for a short srt) and a sufficiently long $T_1$ of the Gd spins (to maintain the polarization induced by the pump pulse during the dipolar evolution time). Broadband excitation pump pulses may alleviate the small modulation depth issue in the swapped setup for spins with large zero field splittings and the acquisition time can be shortened by going to higher temperatures, if possible, or by using

faster relaxing NO labels.

    It would be insightful to perform this type of experiments using a multi-frequency approach to find the best-suited frequency for each DEER channel and spin label combination. However, Q band can be currently considered as the compromise in frequency to perform three-channel DEER experiments with high sensitivity. In fact, GdGd DEER gains in sensitivity by moving to higher frequencies such as W band thanks to a narrowing of the Gd spectrum (Goldfarb, 2014). However, at W

band, NOGd DEER requires dedicated homemade dual mode resonators for an optimal positioning of the pump and observer pulses (Tkach et al., 2011; Kaminker et al., 2013). Additionally, the g anisotropy of the NO spectrum is fully resolved at W band, whereby pump pulses will excite less NO spins, creating lower modulation depths, and, most importantly, orientation selection will have to be taken into account to obtain the correct distance distributions (Polyhach et al., 2007). A large variety of spectroscopically distinguishable spin label pairs is readily available and will be more often used in the future to investigate

complex biomolecular systems owing to the inceased information content that can be obtained from a single sample. Since most spin labels are non-perfectly orthogonal, the methods of identification and suppression of crosstalk signals proposed here can aid to increase DEER signal fidelity in future applications.

*Author contributions.* MT and EB designed the research. MT prepared the EPR samples and performed all EPR experiments. Compound design and synthesis was contributed by MQ, NC, HH, and AG. MT and EB discussed the results and wrote the manuscript. MQ and AG

wrote the SI Part A. The manuscript was revised by all authors.

*Competing interests.* The authors declare no competing interests.

*Acknowledgements.* We acknowledge support by Deutsche Forschungsgemeinschaft (DFG, German Research Foundation) under Germany's Excellence Strategy – EXC-2033 – Projektnummer 390677874 (EB), the DFG Priority Program SPP1601 "New Frontiers in Sensitivity in

EPR Spectroscopy" [DFG BO 3000/2-1 (EB); DFG GO 555/6-2 (AG)], DFG BO 3000/5-1 (EB), DFG INST 130/972-1 FUGG (EB) and SFB958 – Z04 (EB). The Q-band resonator was kindly gifted by G. Jeschke (ETH Zürich/Switzerland).

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
