# Peer review of "Strategies to identify and suppress crosstalk signals in DEER experiments with Gd(III) and nitroxide spin-labeled compounds"

_Magnetic Resonance, 2020_

## Author Comment (AC1) · 1 Jul 2020

Thanks to a private communication with Dr. J. E. Lovett, we became aware of a paper in which a Gd-Gd crosstalk signal was detected in a GdNO DEER channel (Shah et al. Inorg. Chem. 2019, 58, 5, 3015–3025). The authors swapped pump and observer frequency, which is suggested as possible method to suppress the crosstalk signal in our manuscript, and indeed the crosstalk signal decreased. This paper will be cited in the final version of the manuscript.
* * *

---

## Referee Comment (RC1) · Anonymous Referee #1 · 8 Jul 2020

This manuscript focuses on DEER distance measurements between Gd(III) and a nitroxide (NO) radical, often referred to as orthogonal spin labeling. One of the motivations for using such labeling schemes is the ability to carry out selective distance measurements , for example if a biomolecule is labeled with one NO and one Gd(III), then one can probe intra molecular distance via Gd-nitroxide and intermolecular distances (which can arise from oligomerization) by Gd(IIII)-Gd(III) or NO -NO distance measurements. This approach was introduced already in 2012 (DOI: 10.1039/C2CP40282C, Phys. Chem. Chem. Phys., 2012, 14, 10732-10746, which unfortunately is not referenced by the authors). Other reasons maybe increased sensitivity compared to Gd(III)-Gd(III) and elimination of the effect of the dipolar pseudosecular terms on the DEER

modulation frequencies in the case of short distances. In this work the authors used three model compounds with two NO, one NO and one Gd(III) and two Gd(III)-Gd(III) and use them to evaluate how selective are the Gd(III)-Gd(III), Gd(III)-NO and NO-NO distance measurements , while exploiting the different spectral and spin dynamics properties, which have been highlighted in earlier works. This work does not present any new original ideas but using well defined model compounds that can be mixed in a control manner they clearly show expected pitfalls and when they can be overcome and when not. These arise from the spectral overlap of Gd(III) and NO throughout the spectral width of the NO. The authors refer to the consequences of this overlap in various pulse set-ups for DEER as "cross talk" . The value of this manuscript is mainly "educational' as it nicely highlights all issues involved in such measurements on controlled samples. The authors borrowed from optics the nomenclature of color channels to accompany their explanations and in the figs use the associated colors, which again has educational value. I think that after appropriate revisions following the comments below this manuscript will be of value to practitioners of DEER and therefore I recommend publication.

1. In Fig. 3 the bandwidth of the pump and observe pulses are assumed to be the same but I think that this is incorrect, the bandwidth of the an echo detection sequence (two or three pulses) is not the same as just that of the pi pulse. This is even mentioned by the authors (page 13, line 220). Please calculate the correct bandwidth and change Fig. 3.

2. The manuscript is very qualitative and its level can be increased by calculating the predicted modulation depth for NO and Gd(III) at the relevant pump frequencies and compare to the observations. As they have the full lineshape of the Gd(III) and the NO this can be easily done. Similarly, they can account for the degree of overlap for the observe sequence for the different conditions. Such calculations can actually serve to guide the experimental optimized set up. In table S2 the authors mention "Theoretically possible" but as they did not do any theoretical calculations, this term is inappropriate.

3. The presentation of normalized distance distributions without the uncertainties evaluated by validations are misleading. For example in Fig. 1c the trace is very noisy and the modulation depth is small, yet the distance distribution is nice and intense just like the one below. This is just one example but is occurs in many of the Figs. This should be corrected, the P(r) values should be noted on the Y axis and uncertainties should be shown. In Fig. S1 they show that there is no real difference between Gaussian and Tikhonov regularization . So if they chose Tikhonov regularization this maybe easier to show.

4. Why was the Gd(III) pulse taken as 24 ns, when there is enough power to shorten it and improve SNR.

5. Please explain why you choose to add the Gd-Gd ruler in a twice as much concentration, is this to enhance the "cross talk" ?

6. The spectrometer artifact is worrisome – it is larger than the cross talk. What is the source of the artifact and why it appears only in the red channel?

7. P. 3 line 67 : You should use TM (phase memory time) and not T2. Also the differences in phase memory time of Gd(III) and NO is not very different. If you know of cases where it has been used to filter NO and Gd(III) please give a ref.

8. It is more appropriate to cite the original papers than a review. There are not so many examples of Gd(III) –nitroxide distance measurements so better give credit to the original papers and not a review.

9. In general the referencing is rather poor, focusing on self-citations. The omission of the work of Lovett is one example. Another one is the omission of distance measurements between three different spins (Gd(III), nitroxide and Mn(III) (Goldfarb group) and the reference mentioned at the beginning of this evaluation. P. 2 line 35 please give a reference to the DD software as well when mentioning Gaussian fits.

10. Isn't the easiest way the identify the X2 and X3 crosstalk is just running a Gd(III)-

[Figure]

Gd(III) set up and see that it is the same distance as observed in the cross talk.

11. Please shorten the conclusions – no reason to have a two page conclusion that just repeat the results. Should be short and to the point.

---

## Referee Comment (RC2) · Anonymous Referee #2 · 20 Jul 2020

Teucher et al. describe a systematic investigation of distance distribution artifacts that can occur in orthogonally spin-labelled biomacromolecules when specific spins cannot be exclusively addressed but the pulses also excite other spins unintentionally. There are no new concepts or experiment designs in this manuscript but the declared aim is to provide a strategy for identifying and possibly removing 'false positive' distance contributions. While the results do not bring many surprises, this could have been a worked example of how one can thoroughly identify and quantify these artifacts in the distance distribution. However, with the current lack of quantification and error estimation in the analysis this is almost entirely anecdotal with limited value to practitioners. Once the artifacts are quantified and the most important avenues for their suppression

explored and experimental uncertainties are given this may become publishable. In the current state publication would be premature.

Model system

1 You find a much broader distance distribution of the NO-NO ruler than found for the homologous ester-linked structures. As this is unlikely to be rooted in real backbone flexibility (cf. Jeschke JACS 2010 cited in here) and acid amides will not be more flexible than esters either I would suspect a distribution in small exchange couplings. How does the fast motion cw EPR compare with ester-linked rulers? It seems odd to generate a new structure for this study and not fully investigate its spectroscopic properties.

DEER setup

2 You describe the experiments insufficiently to allow independent reproduction of the results. How were the power levels calibrated? You write reducing the AWG output from 100% to 22% corresponds to 12 dB attenuation. Is this frequency independent, is this with the TWT in saturation at 100%? With some of your results not showing the expected microwave power response it is important to understand how the settings were optimized and controlled. Were all pulse power levels optimized via nutation experiments at the respective frequencies to account for the limited resonator bandwidth? This is not clear from the current description. What are the expected differences in nutation between Gd and NO in the NONO and GdNO channels? The contribution from the central transition is not all out dominant at these pump frequencies.

Distance analysis

3 Increase the size of your figure panels. Six-panel wide figures with uniformly scaled distance distributions and DEER signals make it very hard to see the detail of the data. Many of the figures are only 3 panels wide with lots of white space around the data panels.

4 Gaussian fitting does indeed allow a much more stable parametrized analysis. The comparison with Tikhonov Regularization must be extended to the pure rulers (Fig S3). None of the Gaussian fits is particularly good so that the model free analysis has to be shown.

5 Nevertheless, the Gaussian fits allow straightforward quantification of contributions of different distances to the modulation depth. This should be don't throughout and replace the qualitative discussion (see below). The GdGd ruler should contribute to the NONO channel. The signal may be too weak to detect but this should at least be mentioned here. Looking at the spectral overlap a contribution of the GdNO ruler to the NONO DEER does not seem "surprising" at all.

Modulation depths

6 NONO DEER gives 35% and NOGd 30%. You must provide error estimates. Is the difference significant? These are both synthetic rulers with 100% nitroxide labelling. One might assume the modulation depths should be identical unless you can give reasons for the opposite. This needs to be quantitatively addressed.

7 Once the modulations depths are quantified and Gaussian contributions to the distances have been fitted it is straightforward to quantify the contributions of the different spin pairs to the DEER signal in question. This is currently only qualitative (e.g., line 140 "with a slightly smaller modulation depth"). The quantified depths can then be compared with the predictions from the respective modulation depths of the pure rulers. I expect to see a table with the different experiments and samples listing the expected and experimentally found modulation depths and contributions of individual rulers expected and found. Finally, you can add the pure ruler DEER signals in the calculated ratios and show that the contributions are similar to experiment and that the analysis does or does not recover the artifact.

Channels and cross-talk

8 I fail to see the benefit this new nomenclature brings over previous descriptions. There may be some point in the choice of these terms but this should be explained comprehensively as currently it only unnecessarily adds to the confusion. Especially assigning the same distance contributions different cross-talk names whether found with a corresponding spin pair present seems arbitrarily expanding the complexity. What is the added value?

9 According to figures S4 and S5 you only see the GdGd contribution to GdNO experiment in the equimolar samples and no other crosstalk at all. This means doubling the content of GdGd ruler was done to see the other artifacts at all and is biased from the outset. You should be transparent and explicit about this from the outset when describing the setup and results.

10 You derive conclusions from data you refuse to show. This violates basic research transparency and either the data needs to be added or the statements removed (line 193, 233-234)

11 You attribute the GdNO contribution in the NONO experiment to both contributions of Gd to the echo and to the pumped spins. This is based on a 12 dB pump power reduction not altering the modulation depth. How large is the Gd echo at 50K and the chosen refocused echo position? Is it not more likely that the pumping of Gd far off the maximum seems to be invariant to the power levels used in agreement with the data of further experiments (see below)?

12 When reducing the pump power in the GdNO experiment this does not seem to alter the GdGd contribution significantly but the GdNO contribution. You state its distance peak intensity increases but contradict this in the next paragraph by stating its modulation depth reduces. You must quantify the contributions (see above) to make quantitative statements. The statement of "optimized pump power" seems peculiar as the modulation depth reduces with contradicting this more optimum setup. It seems the dependence of the modulation depth on the pump power on Gd away from the
maximum is not understood and largely invariant to pump power if not contradicting the predicted trends. The discussion has to reflect this. The power dependence of the spectra in Fig 8 indicate that none of your spins is experiencing the nominal flip angles at 100%.

Spectrometer-specific artifact

13 You should be able to see this artifact in its pure form using a sample of free Gd and NO spin label. How do you know it is an artifact? How do you know it is spectrometer-specific? How many other instruments with the same nominal configuration have you tried?

GdNO DEER

14 The main potential advantage of NO detected Gd pumped DEER is that 50 K can be used for fast repetition on the nitroxide and diminishing contributions of Gd to the refocused echo as transverse dephasing should be fast. This should definitely be compared experimentally with the other GdNO DEER setup used in here but is not even mentioned. The experiment in Fig 8 done at 50 K will be insightful in first instance. The sentence "...but experimentally impracticable for samples containing NO and Gd spins due to the prohibitively long shot repetition time of the experiment and the small modulation depths expected." in the conclusion should be adapted in the light of this.

Conclusion

15 The conclusion should not repeat the findings at length but conclude the relevant achievement with respect to the state of the art and the resulting implications and several points of discussion should be moved to the relevant section: -GdGd crosstalk in NONO DEER is likely to be diminished by a negligible Gd refocused echo at 50K and this is why the NO detected GdNO DEER and the Gd transverse dephasing at 50 K need to be given for comparison. -The suggestion to produce new samples lacking certain spins to prove crosstalks is directly opposed to this manuscript's aim. If

you make these samples anyways why bother with identifying crosstalks? The GdNO DEER pumping Gd will likely be more cost-effective. -GdGd crosstalk in the GdNO channel can be identified by a minor change in modulation depth upon pump pulse power reduction but if the modulation depth collapses to $\sim$15% how do I exclude the presence of GdGd crosstalk?

Minor

-"The term orthogonal refers to spin labels that are spectroscopically distinguishable from each other and can be addressed and/or detected independently, e.g. via distinct resonance frequencies, relaxation behavior or transition moments." It would be very helpful to readers if at least one example per concept (frequency, relaxation and nutation filtering) could be given rather than none at all.

-In section 1.3 you quantify the spectral widths and relative nutation frequencies but not relaxation differences. You can help the reader by giving longitudinal and transverse magnetization decay constants for both spins at 10 and 50 K to follow this rationale.

-Caption figure 4: "Regions in which distances can be theoretically expected". Outline the theory and how this determines where distances can be expected in practice.

- "Accordingly, we suggest that the dominant signal contribution at 2 nm arising from the NO-NO ruler masks the NO-Gd crosstalk signal." This can easily be checked by synthesizing data from the two pure rulers in the corresponding ratio and analyzing it.

-Figure S1 You seem to observe some orientation correlation in the GdNO ruler, does the small short-distance spike in the Tikhonov distance distribution correspond to double the frequency of the main peak?

The manuscript has a plethora of general statements that need modification or at least significant context:

-You give 8 nm as upper limit for DEER which is half the current maximum claimed in literature.

[Figure]

-Your discussion of background correction relies on a homogeneous distribution of spins. This should at least be mentioned.

-You should clarify the definition of the form factor, when comparing the initial definition by Milov et al. and the more recent use by Jeschke this means different things.

-The multi-spin problem leads to ghost peaks as you rightly state, but it also leads to loss of intensity and resolution at longer distances.

-Your definition of spectroscopically orthogonal seems ambiguous. As it is impossible to independently address the nitroxide it would fall outside the definition of being orthogonal to the Gd.

-Spectral overlap between metal ion and nitroxide is common for Gd, Mn, Fe but not for Cu.

-"Nitroxides (NO) and GdIII-based spin labels (Gd) are the most commonly used orthogonal spins for DEER experiments on biomolecules." Please provide evidence for this statement. The selective citation practice does not back this up.

-"For the Gd-Gd crosstalk signals in the NOGd DEER channel, which are the most relevant unwanted signals in the analysis of complex protein mixtures. . ." There should be evidence provided for this assertion.

-"Q band currently offers the highest sensitivity to perform the three-channel DEER experiments with samples containing both NO and Gd spin labels on a commercial spectrometer." There is justification or references needed for this statement.

---

## Author Response (AR1)

RUHR-UNIVERSITÄT BOCHUM | 44801 Bochum | Germany

**FAKULTÄT FÜR CHEMIE UND BIOCHEMIE**

**AG EPR-Spektroskopie**
Gebäude NBCF 03/498
Universitätsstraße 150, 44801 Bochum

**Prof. Dr. Enrica Bordignon**
Fon  +49 (0)234 32-26239

email : enrica.bordignon@rub.de

To the Editor of Magnetic Resonance

**3. September 2020**

Dear Editor,
Dear Prof. Prisner,

Please find enclosed the revised version of the manuscript entitled: 'Strategies to identify and suppress crosstalk signals in DEER experiments' by Markus Teucher, Mian Qi, Ninive Cati, Henrik Hintz, Adelheid Godt and Enrica Bordignon.

We thank both reviewers for their comments, which we addressed in the revised version.
The main changes are briefly summarized in the following:

1. We now provide a quantification of the crosstalk signals based on the relative intensities of the peaks in the distance distribution (new Table S5 in SI Part B).
2. Concerning the error estimation, we focused on the presence of distance peaks from the rulers present in the sample and did not highlight the uncertainties in the distance distributions. We agree that this should be done. Therefore, we provide now in all DEER figures in the main text (new Figs. 4-9) the distance uncertainties obtained via DeerNet, which we think is nowadays the most suitable approach to reliably obtain such errors.
3. We provide a detailed analysis of the nutation experiments performed to optimize the DEER channels for the different spin types and relaxation data on all rulers at different temperatures (new Fig. 2 and new Section 3 in the SI Part B).
4. We now exploit the potential of the swapped setup (new panel d in Fig. 3) that we previously suggested for the NOGd channel at higher temperatures. We found that 30 K is the best compromise to have a relatively short T1 of the observer nitroxide (fast acquisition time) and a relative long T1 (longer than the dipolar evolution time) of the pumped Gd spins. We found that this setup can suppress the crosstalk signal, although providing lower modulation depth and lower signal-to-noise than the conventional setup. Therefore, in the revised version, we provide not only a quick identification strategy for the Gd-Gd crosstalk signal in the NOGd DEER channel, but also a suppression strategy (new Fig. 9 and new Fig. S3 in the SI Part B).

[Figure]

5. We changed the title to introduce the new suppression strategy and we removed the term 'orthogonal', which, despite being used in literature, is clearly not completely adequate for nitroxide and gadolinium spins, based on the described crosstalk effects.

We think that the revised version increased the quality and clarity of the presented study, and we will be delighted to see this manuscript published in MR.

Finally, we would like to point out that some comments from anonymous reviewer 2 were inappropriate. According to COPE ethical guidelines for peer reviewers, the peer reviewers should be objective and constructive in their reviews, refraining from being hostile or inflammatory and from making libelous or derogatory personal comments.

Sincerely, on behalf of all authors

Prof. Dr. Enrica Bordignon
Ruhr-Universität Bochum
Germany

**Point by point response to the Anonymous Referee #1**

**(in grey the comments of the ref #1, in black our responses)**

This manuscript focuses on DEER distance measurements between Gd(III) and a nitroxide (NO) radical, often referred to as orthogonal spin labeling. One of the motivations for using such labeling schemes is the ability to carry out selective distance measurements, for example if a biomolecule is labeled with one NO and one Gd(III), then one can probe intra molecular distance via Gd-nitroxide and intermolecular distances (which can arise from oligomerization) by Gd(IIII)-Gd(III) or NO -NO distance measurements. This approach was introduced already in 2012 (DOI: 10.1039/C2CP40282C, Phys. Chem. Chem. Phys., 2012, 14, 10732-10746, which unfortunately is not referenced by the authors).

This approach was introduced before by the same pioneering authors (Lueders, Jeschke and Yulikov in 2011, https://doi.org/10.1021/jz200073h) which was already referenced. However, the suggested reference is added together with the first NO-Gd measurements from an independent group (Kaminker, PCCP 2012, DOI: 10.1039/c2cp40219j).

Other reasons maybe increased sensitivity compared to Gd(III)-Gd(III) and elimination of the effect of the dipolar pseudosecular terms on the DEER modulation frequencies in the case of short distances. In this work the authors used three model compounds with two NO, one NO and one Gd(III) and two Gd(III)-Gd(III) and use them to evaluate how selective are the Gd(III)-Gd(III), Gd(III)-NO and NO- NO distance measurements, while exploiting the different spectral and spin dynamics properties, which have been highlighted in earlier works. This work does not present any new original ideas but using well defined model compounds that can be mixed in a control manner they clearly show expected pitfalls and when they can be overcome and when not. These arise from the spectral overlap of Gd(III) and NO throughout the spectral width of the NO. The authors refer to the consequences of this overlap in various pulse set-ups for DEER as "cross talk". The value of this manuscript is mainly "educational' as it nicely highlights all issues involved in such measurements on controlled samples. The authors borrowed from optics the nomenclature of color channels to accompany their explanations and in the figs use the associated colors, which again has educational value. I think that after appropriate revisions following the comments below this manuscript will be of value to practitioners of DEER and therefore I recommend publication.

Thanks for the positive evaluation of our work. We agree that this work is educational, and it is meant to give a clear picture of the pitfalls that can be encountered when working with two non-perfectly orthogonal spin labels. In the revised version we added 'non-perfectly orthogonal' to the title, provided a quantification of the relative modulation depths and the uncertainties in the DEER data, a series of relaxation measurements at different temperatures and we developed an additional strategy to suppress the unwanted Gd-Gd crosstalk signals from the NOGd channel (swapped NOGd DEER setup at 30 K). We borrowed the terms 'channel' and 'crosstalk' from optics to simplify the description of the different DEER setups, and to avoid confusion between crosstalk signals, the already known 2+1 signals and possible artifacts in DEER traces.

1.  In Fig. 3 the bandwidth of the pump and observe pulses are assumed to be the same but I think that this is incorrect, the bandwidth of an echo detection sequence (two or three pulses) is not the same as just that of the pi pulse. This is even mentioned by the authors (page 13, line 220). Please calculate the correct bandwidth and change Fig. 3.

In the setup used in this work we have all observer pulses with the same length (32 ns Gaussian), to avoid different excitation bandwidths by the pi/2 and pi pulses in the observer sequence. The excitation bandwidth is calculated with the Easyspin functions 'exciteprofile' (https://easyspin.org/easyspin/documentation/exciteprofile.html) and 'pulse' (https://easyspin.org/easyspin/documentation/pulse.html), which allow using different pulse shapes. 'exciteprofile' computes the excitation profiles for an input pulse function using two-level density matrix dynamics starting from thermal equilibrium (Mx = 0, My = 0, Mz = 1). The magnetization output cannot be used to continue density propagation, therefore we could not directly calculate the excitation profile of the three Gaussian pulses separated by the given interpulse delays. However, the presented excitation profile of the observer in Fig. 3 was used to aid the positioning of the observer and pump pulses to avoid overlap, and therefore the appearance of the 2+1 signal at the end of the time trace, as done in Teucher and Bordignon, JMR, 2018. In the legend of Fig. 3, we clarified further that the presented excitation profile is  from one ideal Gaussian pi pulse, without taking into account the spectral function, the non-linearity of the signal response, the cavity profile or the Q factor.

2. The manuscript is very qualitative and its level can be increased by calculating the predicted modulation depth for NO and Gd(III) at the relevant pump frequencies and compare to the observations. As they have the full lineshape of the Gd(III) and the NO this can be easily done. Similarly, they can account for the degree of overlap for the observe sequence for the different conditions. Such calculations can actually serve to guide the experimental optimized set up. In table S2 the authors mention "Theoretically possible" but as they did not do any theoretical calculations, this term is inappropriate.

We followed your suggestion: we added a new supplementary table that contains an overview of the modulation depths and the relative populations of the individual distances (see Table S5, SI Part B). Concerning the calculation of expected modulation depths with Gd spins: this is not a trivial task since it was already shown that it is difficult to match the theoretical modulation depths for the GdGd channel with those experimentally detected (see Goldfarb,  PCCP, 2014, 16, 9685–9699 and Manukovsky et al., J. Chem. Phys. 2017, 147 (1–9), No. 044201). Notably the modulation depths of the GdGd channel also depends on temperature and on the length of the traces (Gordon-Grossman et al., PCCP, 13(22):10771-80). Additionally, to take into account the crosstalk signals, we should consider the non-optimal excitation of the spins with different transition moments both in the observer and in the pump pulse. As a last complication, the pump pulse is placed at different positions within the resonator profile depending on the DEER setup (see Fig. 3) which also must be considered. Such calculations are out of the scope of this manuscript, and they would have a rather limited value for the practitioner since the strength of the crosstalk signal vs real signal is dependent on the chosen setup (pulse lengths, power, positioning) and on the relative ratio between the spin types present in the sample, therefore any quantification would be more or less unique for the system under study. We modified the 'theoretically possible' in the table with 'possible' based on the existing overlap between NO and Gd in the pump and/or observer positions.

3. The presentation of normalized distance distributions without the uncertainties evaluated by validations are misleading. For example, in Fig. 1c the trace is very noisy and the modulation depth is small, yet the distance distribution is nice and intense just like the one below. This is just one example but is occurs in many of the Figs. This should be corrected, the P(r) values should be noted on the Y axis and uncertainties should be shown. In Fig. S1 they show that there is no real difference between Gaussian and

Tikhonov regularization. So if they chose Tikhonov regularization this may be easier to show.

Thank you for this suggestion. Indeed, we focused on the presence of distance peaks from the rulers that we added in the sample and did not highlight the uncertainties in the distance distributions, which are however minor for most data presented. Now we provide the distance uncertainties obtained via DeerNet, that we think is nowadays the most suitable approach to reliably obtain such errors. In fact, the outcome of a Tikhonov or Gaussian validation is strongly dependent on the parameters set by the user, whilst the evaluation of the uncertainties in the DEER traces using DeerNet in DeerAnalysis2019 is unbiased. The neural network analysis is shown for all DEER experiments of the main text (Fig. 4 to 9) and we provide an additional figure (Fig. S9, SI Part B) showing the performance of different neural networks in finding crosstalk signals. Furthermore, we added, as suggested, the y-axis titles in the distance distributions. We kept the Gaussian analysis of the DEER data over Tikhonov since fitting the data using Gaussians allows quantification of the ratio between different peaks, which is now presented in the new Table S5 (SI Part B).

4. Why was the Gd(III) pulse taken as 24 ns, when there is enough power to shorten it and improve SNR.

In the NOGd DEER setup (Fig. 3(b)), the pump pulse on the NO was chosen to be a 24 ns Gaussian pulse (note that this is a 10.2 ns FWHM) and all Gaussian observer pulses were set to 32 ns (note that this is 13.6 ns FWHM). Currently, with this setup this is the maximally available power.

Indeed, the 32 ns pump and observer pulses used in the GdGd DEER setup at 100 MHz separation could be shortened, however, to avoid overlap of the pulses, this would require increasing the interpulse separation and moving the observer to a region of lower spectral density. We also did not want to increase the excitation bandwidth of the pump at the maximum of the Gd much further to avoid partial excitation of the NO spectrum (see Fig. 3(c)). This setup is therefore optimal for our spectrometer, and results in a crosstalk-free GdGd channel.

5. Please explain why you choose to add the Gd-Gd ruler in a twice as much concentration, is this to enhance the "cross talk"?

We clarified this issue in the text (e.g. first paragraph in section 3.2). Indeed, we chose two ratios to show the dependence of the appearance of the crosstalk signals on the relative spin concentrations. Different stoichiometric ratios can in fact be encountered in real experiments on biological systems. Most crosstalk signals appear in both stochiometric ratios but we also clarify that when the Gd-Gd ruler is present in excess (two-fold in this case), the crosstalk signal $X_3$ is visible, however if it is present in a 1:1 ratio, it

becomes negligible. The experiments with the three rulers in equimolar concentrations can be found in the SI Part B, Table S6 and Fig. S5-S7.

6. The spectrometer artifact is worrisome – it is larger than the cross talk. What is the source of the artifact and why it appears only in the red channel?

We also find this artifact very worrisome and tried multiple approaches to identify and suppress it. Unfortunately, to date without any success. We added a new Fig. S8 (SI Part B) in which DEER traces were measured on isolated maleimide Gd-DOTA labels in solution to show that this artifact is a sinusoidal oscillation which is mostly present in the imaginary part of the signal, it is independent of the dipolar frequency, it is not an ESEEM signal, and can be found also in solutions of $MnCl_2$, therefore Gd-independent.

7. P. 3 line 67: You should use TM (phase memory time) and not T2. Also the differences in phase memory time of Gd(III) and NO is not very different. If you know of cases where it has been used to filter NO and Gd(III) please give a ref.

Thank you for this remark. Indeed, the difference in $T_m$ is not sufficient to filter NO and Gd, only $T_1$ can be used. Therefore, we removed this part of the sentence. Additionally, in the revised version of the paper we show a series of data on $T_m$ and $T_1$ measured on different samples, at different spectral positions and at different temperatures (see new section 3 in SI Part B).

8. It is more appropriate to cite the original papers than a review. There are not so many examples of Gd(III) –nitroxide distance measurements so better give credit to the original papers and not a review.

We agree with your suggestion, therefore we added to our knowledge all references about DEER performed on different spin types.

9. In general the referencing is rather poor, focusing on self-citations. The omission of the work of Lovett is one example. Another one is the omission of distance measurements between three different spins (Gd(III), nitroxide and Mn(III) (Goldfarb group) and the reference mentioned at the beginning of this evaluation. P. 2 line 35 please give a reference to the DD software as well when mentioning Gaussian fits.

We had 5 references from our group over 45 references in total. However, we added more references (in particular, see section 1.2). The total number of references is now 65. The DD software is also cited among the possible methods to extract distance distributions (Stein et al., 2015).

10. Isn't the easiest way the identify the X2 and X3 crosstalk is just running a Gd(III)-Gd(III) set up and see that it is the same distance as observed in the cross talk.

Of course, we mention this possibility for the NO-Gd crosstalk signal in the NONO channel ($X_1$) but this strategy can be ambiguous on real biological systems. In this case it is straightforward, because we know which distances to expect in our samples, but this is not a situation generally found, and two channels can have very similar distance distributions.

11. Please shorten the conclusions – no reason to have a two-page conclusion that just repeat the results. Should be short and to the point.

We shortened the conclusions as suggested.

**Point by point response to the Anonymous Referee #2**

**(in grey the comments of the ref #2, in black our responses)**

Teucher et al. describe a systematic investigation of distance distribution artifacts that can occur in orthogonally spin-labelled biomacromolecules when specific spins cannot be exclusively addressed but the pulses also excite other spins unintentionally. There are no new concepts or experiment designs in this manuscript but the declared aim is to provide a strategy for identifying and possibly removing 'false positive' distance contributions. While the results do not bring many surprises, this could have been a worked example of how one can thoroughly identify and quantify these artifacts in the distance distribution. However, with the current lack of quantification and error estimation in the analysis this is almost entirely anecdotal with limited value to practitioners.

Once the artifacts are quantified and the most important avenues for their suppression explored and experimental uncertainties are given this may become publishable. In the current state publication would be premature.

We now provide a quantification of the crosstalk signals based on the relative intensities of the peaks in the distance distributions (see new Table S5, SI Part B) which is of course reflecting the given values of the modulation depths. We would like to highlight that such a quantification may be of limited value for practitioners as it is strongly dependent on the setup and on the samples used, therefore these numbers are more or less unique for the system under study.

Concerning the error estimation, we focused on the presence of distance peaks from the rulers present in the sample and did not highlight the uncertainties in the distance distributions. We agree that this should be done, and we provide now in all DEER figures in the main text (Fig. 4 to 9) uncertainties obtained via DeerNet, which we think is nowadays the most suitable approach to reliably obtain such errors.

We now exploit the potential of the swapped setup that we suggested in the NOGd channel at higher temperatures and found that indeed this setup can suppress the crosstalk signal. Therefore, we have not only identification but also suppression strategies in the new Fig. 9.

Model system

1 You find a much broader distance distribution of the NO-NO ruler than found for the homologous ester-inked structures. As this is unlikely to be rooted in real backbone flexibility (cf. Jeschke JACS 2010 cited in here) and acid amides will not be more flexible than esters either I would suspect a distribution in small exchange couplings. How does the fast motion cw EPR compare with ester-linked rulers? It seems odd to generate a new structure for this study and not fully investigate its spectroscopic properties.

Indeed, the NO-NO-ruler is a new compound, which was chosen because of its very small spin-spin distance which does not overlap with the mean distances of the other two available rulers. The reason for a smaller distance and a broader spin-spin distance distribution of the NO-NO ruler in comparison to structurally related NO-NO rulers with ester instead of amide linkages between the spacer and the nitroxide moiety is a consequence of the side chains at the N atoms. The experimentally found distance

agrees with the nitroxide moiety and the benzene ring being preferentially oriented trans with respect to the NH-CO bond (s-trans conformation). Adding a PEG chain to the N atom lifts this sterically founded preference and thus makes conformations with smaller dihedral angles accessible and energetically comparable. This reduces the spin-spin distance and broadens the distance distribution. Here, we are only interested in having at disposal a distinguishable short distance arising from a NO-NO ruler, however, a detailed analysis of the properties of a series of new rulers will be the subject of another manuscript.

DEER setup

2 You describe the experiments insufficiently to allow independent reproduction of the results. How were the power levels calibrated? You write reducing the AWG output from 100% to 22% corresponds to 12 dB attenuation. Is this frequency independent, is this with the TWT in saturation at 100%? With some of your results not showing the expected microwave power response it is important to understand how the settings were optimized and controlled. Were all pulse power levels optimized via nutation experiments at the respective frequencies to account for the limited resonator bandwidth? This is not clear from the current description. What are the expected differences in nutation between Gd and NO in the NONO and GdNO channels? The contribution from the central transition is not all out dominant at these pump frequencies.

We tried to describe the DEER experiments in a very detailed way as usual, including pulse lengths, types, positions with respect to the spectra, frequency separation, temperature, etc. We wrote 'The length of the pulses was optimized via transient nutation experiments for each spin type as shown in Fig. 2.' We did not describe deeply the standard procedure we use to set up the experiments via nutation, but the AWG amplitudes and shot repetition times shown in the old Fig. 2 should have exemplified this approach. Of course these numbers were not used as general values across all setups, in fact, as written, we optimized the pump and observer pulse lengths for each DEER setup, on each sample and for each spin type individually via transient nutation experiments to account for the different positions in the resonator dip and the limited resonator bandwidth.
We further clarified this aspect in the text by the addition of a new paragraph on nutation experiments (section 2.2.2) and an extension of Fig. 2 (see also Table S2, SI Part B). Fig. 2 shows now the changes in the Gd and NO nutation frequencies at different positions in the spectrum as a function of the main attenuator and the AWG amplitudes. These data illustrate also that the nutation frequencies were very similar when detected at the maximum of the Gd signal or 280 MHz higher in frequency, which indicates that the major contribution arises from the -1/2 to +1/2 transition.

Distance analysis

3 Increase the size of your figure panels. Six-panel wide figures with uniformly scaled distance distributions and DEER signals make it very hard to see the detail of the data. Many of the figures are only 3 panels wide with lots of white space around the data panels.

Thank you for this suggestion, we increased the figure and font sizes and added new panels with the DeerNet analysis.

4 Gaussian fitting does indeed allow a much more stable parametrized analysis. The comparison with Tikhonov Regularization must be extended to the pure rulers (Fig S3). None of the Gaussian fits is particularly good so that the model free analysis has to be shown.

The Tikhonov analysis was already shown for the pure rulers (in the suppl. Figure which is now Fig. S4) to compare the distance distributions with those obtained by Gaussian fit, how could we extend that?

We decided to add a model-free analysis using DeerNet in the main figures which also provides an uncertainty estimate to be compared with the Gaussian analysis. The results from both methods are in good agreement with each other. We kept the Gaussian analysis in the main figures since it allows for an extraction of the relative ratios of the different peaks in the distance distributions, which are now shown in the new Table S5 (SI Part B). Notably, the validation procedure is not available for the Gaussian analysis in DeerAnalysis2019.

5 Nevertheless, the Gaussian fits allow straightforward quantification of contributions of different distances to the modulation depth. This should be don't throughout and replace the qualitative discussion (see below). The GdGd ruler should contribute to the NONO channel. The signal may be too weak to detect but this should at least be mentioned here. Looking at the spectral overlap a contribution of the GdNO ruler to the NONO DEER does not seem "surprising" at all.

A quantification of the distance contributions based on the modulation depths was already present in the text. However, we now added a quantification based on the Gaussian fits which reflect the contributions of the modulation depths. A new table with all extracted parameters is now presented in the SI Part B (Table S5).

We already mentioned that a GdGd crosstalk signal in the NONO DEER channel was possible and classified it as $X_4$ (page 14 line 252 in the submitted manuscript and Table S2 in the original manuscript, now Table S7). We also discussed that it is probably too weak to be detected.

Modulation depths

6 NONO DEER gives 35% and NOGd 30%. You must provide error estimates. Is the difference significant? These are both synthetic rulers with 100% nitroxide labelling. One might assume the modulation depths should be identical unless you can give reasons for the opposite. This needs to be quantitatively addressed.

All mod. depths are now presented in Table S5 (SI Part B). When comparing the data presented in Fig. 4(c) and in Fig. S5, one can see that for an independent repetition (different sample) the modulation depth of the NO-Gd ruler is perfectly reproduced with the same setup, therefore, this uncertainty is minimal. The concentration of the rulers was kept quite low to minimize the effects of the background fit on the modulation depth. However, we show now in Fig. R1 the changes in the modulation depth of the NO-Gd ruler in the NOGd channel by varying the power of the pi pulse from that optimized by nutation experiments, and in Fig. R2 the changes in modulation depth by changing the position of the pump pulse from being in the center of the dip to being positioned exactly in the same position in the dip as for the NONO channel.

[Figure]

**Figure R1:** Effect of changing the pump pulse power (AWG amplitude) on the modulation depth in a NOGd DEER setup with a 32 ns pump pulse placed in the center of the dip using the NO-Gd ruler sample. The power optimized via nutation experiments is 65% (26% mod. depth). Interestingly, at higher power, the modulation depth increases (at 85% power it is 30%, and at 100% power it reaches 32%).

[Figure]

**Figure R2:** Comparison of different NOGd DEER setups using the NO-Gd ruler sample. Shown in red is the NOGd DEER used in the main text which uses a 24 ns Gaussian pump pulse placed in the center of the resonator dip about 0.4 mT (11 MHz) higher in field than the maximum of the NO spectrum. Show in black is the same setup with the pump pulse placed on the maximum of the NO spectrum (its excitation bandwidth extends over low field side of the spectrum, therefore the mod. depth is a bit smaller). Depicted in gray is the same setup with the pump pulse placed at the same position in the dip as in case of the NONO DEER (about 50 MHz higher in frequency than the center of the dip). The largest deviation between the modulation depths is 2%.

Indeed, the modulation depth of the NOGd channel with the same pump as the NONO channel is slightly smaller than that obtained with the NONO channel. The origin of this effect cannot be a nitroxide radical content <100% in the Gd-NO ruler, therefore effects related to the reduction of the observer echo intensity by the pump might be the reason for the decreased modulation depth detected. This possibility was already addressed as possible source of discrepancies in Shah et al., Inorg. Chem. 2019, citing papers related to the echo reduction effects (Gmeiner et al., PCCP, 2017; Kaminker et al., PCCP, 2012; Yulikov et al., PCCP 2012).

When additional Gd-Gd rulers are present in the sample together with the NO-Gd ruler (in the binary and ternary mixture, the modulation depth decreases reproducibly from 30% to 12.5%, due to the fraction of unmodulated Gd signal which is not dipolarly coupled to the nitroxide moiety. We expected a 6% modulation due to the unmodulated 2xGd-Gd signal intensity in the observer echo, which is lower than the value experimentally obtained. The reason for this discrepancy is unknown, however it could also be related to the observer echo reduction in the presence of the pump pulse, which can affect differently the Gd-Gd rulers and the Gd-NO rulers.

7 Once the modulations depths are quantified and Gaussian contributions to the distances have been fitted it is straightforward to quantify the contributions of the different spin pairs to the DEER signal in question. This is currently only qualitative (e.g., line 140 "with a slightly smaller modulation depth"). The quantified depths can then be compared with the predictions from the respective modulation depths of the pure rulers. I expect to see a table with the different experiments and samples listing the expected and experimentally found modulation depths and contributions of individual rulers expected and found. Finally, you can add the pure ruler DEER signals in the calculated ratios and show that the contributions are similar to experiment and that the analysis does or does not recover the artifact.

The table that you expect to see is the new Table S5 (SI Part B). The ratio of the Gaussian peaks represents the ratio of the respective modulation depths.

Channels and cross-talk

8 I fail to see the benefit this new nomenclature brings over previous descriptions. There may be some point in the choice of these terms but this should be explained comprehensively as currently it only unnecessarily adds to the confusion. Especially assigning the same distance contributions different cross-talk names whether found with a corresponding spin pair present seems arbitrarily expanding the complexity. What is the added value?

Reviewer 1 find that borrowing the nomenclature channel and crosstalk from optics increases the generality, and we agree with his comment. By 'nomenclature' the referee means: the DEER 'channels', the 'crosstalk' signals or the 'name' of the crosstalk signals? We think that using 'crosstalk' signal simplifies the description of the observed peak (it is a signal, not an artifact, and it is due to the spectral overlap which makes it impossible to have clean DEER channels; it should be confused with the '2+1' signal present in DEER if rectangular pulses are used), and it is already used to describe similar effects in microscopy. Added value is clarity, and the compact form $X_i$ simplify the description of the crosstalk signals. We used $X_2$ and $X_3$ for the same crosstalk in the presence and absence of a real NOGd signal because it helps the reader to see the differences in the identification and suppression strategies, as explicitly stated in the text.

9 According to figures S4 and S5 you only see the GdGd contribution to GdNO experiment in the equimolar samples and no other crosstalk at all. This means doubling the content of GdGd ruler was done to see the other artifacts at all and is biased from the outset. You should be transparent and explicit about this from the outset when describing the setup and results.

**We would like to point out that we always aim to be transparent in our description, however we might have not reached enough clarity. Transparency is a necessary requirement in science.**

We chose this ratio for the figures in the main text to highlight the appearance of crosstalk signals. We think that a two-fold excess of the Gd-Gd ruler is still a ratio that can easily occur when studying biological systems. For comparison, we performed the same series of experiments with the three rulers in equimolar concentrations, which can be found in the SI Part B, Table S6 and Fig. S5-S7. We highlighted this more clearly (e.g. first paragraph in section 3.2).

10 You derive conclusions from data you refuse to show. This violates basic research transparency and either the data needs to be added or the statements removed (line 193, 233-234)

**This comment is not appropriate. We do not refuse to show data, and we do not violate basic research transparency.**

The policy of the journal does not state that 'data not shown' cannot be used. We are very happy to show all data, although especially nutation experiments are never shown in publications. If the referee would like to see other data, we will be happy to provide any information. As discussed in our answer to point 2, we extended Fig. 2 to include more data (see also Table S2, SI Part B).

11 You attribute the GdNO contribution in the NONO experiment to both contributions of Gd to the echo and to the pumped spins. This is based on a 12 dB pump power reduction not altering the modulation depth. How large is the Gd echo at 50K and the chosen refocused echo position? Is it not more likely that the pumping of Gd far off the maximum seems to be invariant to the power levels used in agreement with the data of further experiments (see below)?

We removed this comment, because indeed the minor change observed does not provide enough info.

12 When reducing the pump power in the GdNO experiment this does not seem to alter the GdGd contribution significantly but the GdNO contribution. You state its distance peak intensity increases but contradict this in the next paragraph by stating its modulation depth reduces.

The overall modulation depth reduces, and the relative intensity of the Gd peak increases, this is correct.

You must quantify the contributions (see above) to make quantitative statements.

All modulation depths are now presented in Table S5 (SI Part B).

The statement of "optimized pump power" seems peculiar as the modulation depth reduces with contradicting this more optimum setup. It seems the dependence of the modulation depth on the pump power on Gd away from the maximum is not understood and largely invariant to pump power if not contradicting the predicted trends. The discussion has to reflect this. The power dependence of the spectra in Fig 8 indicate that none of your spins is experiencing the nominal flip angles at 100%.

We agree that the power dependence of the GdGd channel is difficult to interpret. Optimized means 'by nutation'. Below you find the effects of different pump powers on the GdGd DEER modulation depth (Fig. R3) and on the NONO DEER modulation depth (Fig. R4). In the NONO DEER the modulation depths change as expected, with the maximum being at the optimal pi pump obtained via nutation. In the GdGd channel the behavior is completely different. We cannot explain the deviation observed in the GdGd channel with respect to the expected behavior, but it could be related to the reduction in the observer echo at increasing pump powers (see increasing noise levels in the time traces), and is further complicated by the spectrometer artifact at 3.5 nm. Obviously, the chosen power of the pump pi-pulse optimized by nutation experiments does not provide the maximum modulation depth in GdGd DEER.

[Figure]

Figure R3: Dependence of the modulation depth on the pump pulse power (AWG amplitude) at 0 dB for a GdGd DEER at 100 MHz separation between the pump and observer frequencies. All pulses were set to 32 ns. All time traces are single scans and just differ by the utilized pump pulse power. The asterisk denotes the spectrometer artifact. The pulse lengths were optimized via nutation experiments. 19% pump pulse amplitude corresponds to a pi-pulse on the spectral maximum of the Gd. The relative intensity of the artifact increases with increasing pump pulses power further complicating the analysis of the modulation depth.

[Figure]

(a) pump pulse amplitude variation       (b) pump pulse amplitude variation (selection)

Figure R4: Dependence of the modulation depth on the pump pulse power (AWG amplitude) at 0 dB for a NONO DEER at 100 MHz separation between the pump and observer frequencies. All pulses were set to 64 ns. All time traces are single scans and just differ by the utilized pump pulse power. The pulse lengths were optimized via nutation experiments. 28% pump pulse amplitude corresponds to a pi-pulse on the spectral maximum of the NO.

Spectrometer-specific artifact

13 You should be able to see this artifact in its pure form using a sample of free Gd and NO spin label. How do you know it is an artifact? How do you know it is spectrometer specific? How many other instruments with the same nominal configuration have you tried?

We added a new Fig. S8 (SI Part B) in which DEER time traces were measured on isolated maleimide Gd-DOTA labels in solution to show that this artifact is a sinusoidal oscillation, it is mostly present in the imaginary part of the signal (but clearly it appears also in the real part in some experiments), it is independent of the dipolar frequency, it is not an ESEEM signal, and can be found also in solutions of $MnCl_2$.

GdNO DEER

14 The main potential advantage of NO detected Gd pumped DEER is that 50 K can be used for fast repetition on the nitroxide and diminishing contributions of Gd to the refocused echo as transverse dephasing should be fast. This should definitely be compared experimentally with the other GdNO DEER setup used in here but is not even mentioned. The experiment in Fig 8 done at 50 K will be insightful in first instance.

The sentence "...but experimentally impracticable for samples containing NO and Gd spins due to the prohibitively long shot repetition time of the experiment and the small modulation depths expected." in the conclusion should be adapted in the light of this.

We thank the reviewer for this good idea, indeed we did not try to go higher in temperature. It is tempting to use 50 K because one can use the fast srt due to the fast $T_1$ of the nitroxide. However, at 50 K the $T_1$ of the Gd is in the low microsecond time range (see Table S3, SI Part B), therefore the longitudinal relaxation counteracts the inversion induced by the pump pulse within the dipolar evolution time, which is incompatible with DEER. However, we analyzed the relaxation times at different temperatures (see Fig. S3, SI Part B), and found that 30 K was suitable for DEER because the $T_1$ of the Gd spins is slightly larger than the dipolar evolution time, and we could use an srt of 10 ms (NO observer), detecting DEER traces with a maximal length 2 and 3 microseconds with a good SNR. The swapped NOGd setup allows removing the crosstalk signal, as shown for the $X_2$ and $X_3$ cases in new Fig. 9. In the previous version of the paper, we tried only the swapped experiment at 10 K, with an srt of 100 ms, which indeed was impractical due to the too long acquisition time needed. All data are shown in the new Figure 9.

Conclusion

15 The conclusion should not repeat the findings at length but conclude the relevant achievement with respect to the state of the art and the resulting implications and several points of discussion should be moved to the relevant section: -GdGd crosstalk in NONO DEER is likely to be diminished by a negligible Gd refocused echo at 50K and this is why the NO detected GdNO DEER and the Gd transverse dephasing at 50 K need to be given for comparison. -The suggestion to produce new samples lacking certain spins to prove crosstalks is directly opposed to this manuscript's aim. If you make these samples anyways why bother with identifying crosstalks? The GdNO DEER pumping Gd will likely be more cost-effective. -GdGd crosstalk in the GdNO channel can be identified by a minor change in modulation depth upon pump pulse power reduction but if the modulation depth collapses to ~15% how do I exclude the presence of GdGd crosstalk?

We cut the conclusions. Now that we found a good way to suppress the Gd-Gd crosstalk signal in the NOGd DEER channel, we suggest to produce a new sample only if a signal is present in the NONO DEER channel, which has a distance similar to that obtained in the GdGd channel. If a biological complex mixture of proteins is studied, there might be overlapping real and crosstalk distances, which may complicate the assignment. In case of doubt, before interpreting data erroneously, we believe it is wise to prepare a new sample.

Minor

-"The term orthogonal refers to spin labels that are spectroscopically distinguishable from each other and can be addressed and/or detected independently, e.g. via distinct resonance frequencies, relaxation behavior or transition moments." It would be very helpful to readers if at least one example per concept (frequency, relaxation and nutation filtering) could be given rather than none at all.

The term "orthogonal" is commonly used in literature, we did not introduce it here. We agree that Gd and NO are not perfectly orthogonal, but distinguishable, as shown here. This issue is only a matter of very careful wording and we do not intent to overstretch the meaning of "orthogonal". We changed the title, to account for that.

-In section 1.3 you quantify the spectral widths and relative nutation frequencies but not relaxation differences. You can help the reader by giving longitudinal and transverse magnetization decay constants for both spins at 10 and 50 K to follow this rationale.

We added relaxation data to the manuscript: Table S3-S4 and Fig. S1-S3 (SI PartB).

-Caption figure 4: "Regions in which distances can be theoretically expected". Outline the theory and how this determines where distances can be expected in practice.

Theoretically in that context means 'in principle'. If an overlap exists, one can expect a crosstalk signal. We changed the term 'theoretically possible' to 'possible' (see Fig. 4 and Table S7, SI Part B).

- "Accordingly, we suggest that the dominant signal contribution at 2 nm arising from the NO-NO ruler masks the NO-Gd crosstalk signal." This can easily be checked by synthesizing data from the two pure rulers in the corresponding ratio and analyzing it.

We now provide a relaxation-based rationale for not seeing that artifact when NO signals are present and additionally, we address the excess of NO signal in the echo, which further diminishes the relevance of the crosstalk if NO signals are present.

-Figure S1 You seem to observe some orientation correlation in the GdNO ruler, does the small short-distance spike in the Tikhonov distance distribution correspond to double the frequency of the main peak?

DeerNet does not pick up this small spike (see Fig. 4(a)), therefore we do not consider it in the analysis. Even if there is a minor orientation selection it does not influence the data interpretation, however, we now added this info in the text.

The manuscript has a plethora of general statements that need modification or at least significant context:

-You give 8 nm as upper limit for DEER which is half the current maximum claimed in literature.

We are aware of this publication but we decided to state 8 nm as an upper distance limit for DEER experiments since we believe that 16 nm is not within practically reachable limits considering that it requires a perdeuterated sample in a perdeuterated buffer - i.e. conditions hardly used in structural biology. However, we rephrased the sentence and added this reference (page 1 line 15).

-Your discussion of background correction relies on a homogeneous distribution of spins. This should at least be mentioned.

We added this information (see page 6, line 146). However, due to the low spin concentration, the same results are obtained by 2D or 3D background correction.

-You should clarify the definition of the form factor, when comparing the initial definition by Milov et al. and the more recent use by Jeschke this means different things.

The Form factor is the primary DEER trace divided by the background function and normalized to 1. Added to the first figure legend showing an F(t).

-The multi-spin problem leads to ghost peaks as you rightly state, but it also leads to loss of intensity and resolution at longer distances.

Yes, we added this extra information (see page 2, line 42f).

-Your definition of spectroscopically orthogonal seems ambiguous. As it is impossible to independently address the nitroxide it would fall outside the definition of being orthogonal to the Gd.

This term is used in literature. See comment before on the term 'orthogonal'. Now in the revised text we make it clear that Gd and NO are non-perfectly orthogonal.

-Spectral overlap between metal ion and nitroxide is common for Gd, Mn, Fe but not for Cu.

This is true, our statement was too general. We removed it from the text.

-"Nitroxides (NO) and GdIII-based spin labels (Gd) are the most commonly used orthogonal spins for DEER experiments on biomolecules." Please provide evidence for this statement. The selective citation practice does not back this up.

We included in the last paragraph of section 1.2 all literature to our knowledge that can be found on NO in conjunction with other spin labels. The amount of publications on NO-Gd-labeled systems strongly supports our hypothesis.

-"For the Gd-Gd crosstalk signals in the NOGd DEER channel, which are the most relevant unwanted signals in the analysis of complex protein mixtures..." There should be evidence provided for this assertion.

This is the output of our comparative analysis. The Gd-Gd crosstalk are the most disturbing and appear in the NOGd DEER channel.

-"Q band currently offers the highest sensitivity to perform the three-channel DEER experiments with samples containing both NO and Gd spin labels on a commercial spectrometer." There is justification or references needed for this statement.

Justification is provided by discussing the advantages and disadvantages in going to lower and higher frequencies.

[revised manuscript text omitted]

---

## Editor Decision (ED1)

Dear Authors of the manuscript ***Strategies to identify and suppress crosstalk signals in DEER experiments***

After careful reading your revised manuscript (including the revised Supporting Information), your detailed answers to the initial comments of both reviewers and the new comments of both reviewers I decided that your work will be finally accepted as article in Magnetic Resonance if you make the following minor revisions on your manuscript (based on the following objective and constructive remarks and suggestions of both reviewers):

1) Please change the title of your manuscript to: *Strategies to identify and suppress crosstalk signals in DEER experiments of Gd(III)-nitroxide spin labeled samples.*
2) Please correct Figure 2B 4$^{th}$ column accordingly
3) Please add in the SI in chapter B. 3.1. an additional Figures showing one representative T1 time traces for NO and for Gd(III).
4) Please rename chapter B. 3.2. in the SI to *Translational relaxation time* and define the shown values in Figure S2 as *Tm10%*
5) Please add a sentence of the expected linearity of AWG percentage to final output microwave power after amplification somewhere in the text.

All other comments made by reviewer 2 are to my opinion not appropriate and have not to be addressed in the final version of the manuscript (or are already satisfactorily addressed in the revised version of the manuscript).

---

## Author Response (AR2)

**Reply to editor's comments**

**1) Please change the title of your manuscript to: Strategies to identify and suppress crosstalk**

**signals in DEER experiments of Gd(III)-nitroxide spin labeled samples.**

The title was changed accordingly to "Strategies to identify and suppress crosstalk

signals in DEER experiments with Gd(III) and nitroxide spin-labeled compounds"

**2) Please correct Figure 2B 4 th column accordingly**

We changed the legend of this figure to make clearer which points were measured at which power.

**3) Please add in the SI in chapter B. 3.1. an additional Figures showing one representative T1**

**time traces for NO and for Gd(III).**

We added the requested plots to Fig. S1 and renamed the relaxation times according to our utilized method as "$T_{1\,[0.26]}$".

**4) Please rename chapter B. 3.2. in the SI to Translational relaxation time and define the shown**

**values in Figure S2 as Tm10%**

Done, we also added example plots to illustrate the data evaluation and renamed the relaxation times according to our utilized method as "$T_{m\,[10\%]}$".

**5) Please add a sentence of the expected linearity of AWG percentage to final output microwave power after amplification somewhere in the text.**

[Figure]

A sentence was added: The linear dependence between the AWG amplitude and the intensity of the transiently recorded pulses in transmission mode (TM) was previously shown (SI of Teucher and Bordignon, 2018), and here we demonstrate that this linearity is maintained also after the TWT (Traveling-Wave Tube) amplification up to 70-80% AWG amplitude (at this input power the TWT starts to saturate).

The plot shown here is the dependency of the inverse of the pi pulse length (proportional to B1) after the TWT amplification- vs the AWG input amplitude (using the data from Figure 2).

**Referee #1 point by point answers**

**Please add to the title "Gd(III)-nitroxide", the title now is too general. In the response the authors noted that they added "'non-perfectly orthogonal' to the title ", I do not see this in the revised version with track changes. In any case it is no proper.**

Thank you for this suggestion. We changed the title accordingly. "Non-perfectly orthogonal" was a leftover from an earlier revised version of the title.

**Give a ref for the statement "A reliable fit of the background function relies on recording the primary DEER time trace as long as possible, so that the last 2/3 of the trace contains a pure background decay function."**

We rephrased that sentence.

**Fig. 2B, right column, I see only the trace of 12 dB, though 6 and 0 are noted on the Fig. as well.**

Indeed, we do not show the full traces for 6 and 0 dB but we show some single point for both powers. We changed the legend to make clearer which points were measured at which power.

**The addition of the data analysis using DEERnet is welcome and serves as a good benchmark as compared to the Gaussian fit. DeerNet analysis is based on a training set of NO-NO DEER, yet you are using it also for Gd-NO and Gd-Gd. Maybe this deserves a comment?**

This is a good point. We added a remark about this in section 2.2.3 where we introduce the utilized DEER data evaluation methods. Indeed, Deernet performs well with GdGd and NOGd DEER.

**Fig. S4 is mentioned in the text before S1, S2 and S3. Shouldn't this be consecutive.**

Fig. S1 to S3 are first mentioned in the legend of Fig. 2 on page 5, while Fig. S4 is mentioned the first time on page 6 in the main text. – Accordingly, the SI figures are mentioned consecutively in reading order.

**Fig. 3 – are the spectra a superposition of the NO and Gd(II) spectra, or a measurements of the Gd(III)-NO ruler. Please make this clear.**

We changed the first sentence in the legend of Fig. 3 to make clearer that the spectra are superimposed.

**T1 measurements - I am surprised that the recovery is exponential, usually it is not and requires either a stretched exponential or two exponents. Can you please show in the SI examples of two traces and their fit (one for NO and one for Gd(III)).**

We added the requested plots to Fig. S1. Indeed, the data cannot be satisfyingly fitted using a monoexponential fit but requires either a stretched exponential (as shown now in Fig. S1) or a biexponential fit function. We decided to use the time T at which the inversion recovery echo signal has an intensity of 0.26 since this is the method which requires the least number of parameters and can provide a qualitative longitudinal relaxation value that can be used to compare different samples. We did not aim to determine absolute T1 values (if this is possible at all). To account for the unsurprisingly different relaxation times obtained using a stretched exponential and the 0.26 method we used subscripts ($T_{1\,[exp]}$ or $T_{1\,[0.26]}$).

**Naming the time the echo decays to 10% of its value as TM is unfortunate, because TM is defined as the phase memory time, which is obtained from a fit to an exponent decay, or a stretched exponential. Better use T10%.**

Agreed. We added some example traces in Fig. S2 and renamed the relaxation times as "$T_{m\,[10\%]}$" indicating the utilized method to extract the value. Again, we decided to use this method to have the least number of parameters for the fit and to be able to provide a qualitative comparison between different samples.

**It will be fair to acknowledge at the conclusions and outlook that the cross-talk effect is rather small. Currently, peaks with 10% intensity of the main peak in the distance distribution are ignored.**

We do mention in the conclusions and outlook that the signals are in the order of 10% of the maximally expected modulation depth but we also mention that these numbers are only valid for the respective molar ratio of the utilized samples. We show that the strength of the crosstalk signals is dependent on the relative molar ratio between the rulers (1:1:2 in the main text versus 1:1:1 in the SI). By changing the molar ratio between the labels and depending on the distances present in the sample, crosstalk signals can also become a much more dominant contribution in a DEER channel.

We also do not think that the signals could be generally ignored during data evaluation, especially if, as in the case of the crosstalk signals X1 and X2, they are the only and thus dominant contribution in the respective DEER channels.

[revised manuscript text omitted]